# Reshuffling Resampling Splits Can Improve Generalization of Hyperparameter Optimization

**Thomas Nagler**[*]
t.nagler@lmu.de

**Lennart Schneider**[*]

**Bernd Bischl**

**Matthias Feurer**

Department of Statistics, LMU Munich
Munich Center for Machine Learning (MCML)

## Abstract

Hyperparameter optimization is crucial for obtaining peak performance of machine learning models. The standard protocol evaluates various hyperparameter configurations using a resampling estimate of the generalization error to guide optimization and select a final hyperparameter configuration. Without much evidence, paired resampling splits, i.e., either a fixed train-validation split or a fixed cross-validation scheme, are often recommended. We show that, surprisingly, reshuffling the splits for every configuration often improves the final model's generalization performance on unseen data. Our theoretical analysis explains how reshuffling affects the asymptotic behavior of the validation loss surface and provides a bound on the expected regret in the limiting regime. This bound connects the potential benefits of reshuffling to the signal and noise characteristics of the underlying optimization problem. We confirm our theoretical results in a controlled simulation study and demonstrate the practical usefulness of reshuffling in a large-scale, realistic hyperparameter optimization experiment. While reshuffling leads to test performances that are competitive with using fixed splits, it drastically improves results for a single train-validation holdout protocol and can often make holdout become competitive with standard CV while being computationally cheaper.

## 1 Introduction

Hyperparameters have been shown to strongly influence the performance of machine learning models (van Rijn & Hutter, 2018; Probst et al., 2019). The primary goal of hyperparameter optimization (HPO; also called tuning) is the identification and selection of a hyperparameter configuration (HPC) that minimizes the estimated generalization error (Feurer & Hutter, 2019; Bischl et al., 2023). Typically, this task is challenged by the absence of a closed-form mathematical description of the objective function, the unavailability of an analytic gradient, and the large cost to evaluate HPCs, categorizing HPO as a noisy, black-box optimization problem. An HPC is evaluated via resampling, such as a holdout split or $M$-fold cross-validation (CV), during tuning.

These resampling splits are usually constructed in a fixed and instantiated manner, i.e., the same training and validation splits are used for the internal evaluation of all configurations. On the one hand, this is an intuitive approach, as it should facilitate a fair comparison between HPCs and reduce the variance in the comparison.[1] On the other hand, such a fixing of train and validation splits might steer the optimization, especially after a substantial budget of evaluations, towards favoring HPCs

---

[*]Equal contribution.

[1]This approach likely originates from the concept of paired statistical tests and the resulting variance reduction, but in our literature search we did not find any references discussing this in the context of HPO. For example, when comparing the performance of two classifiers on one dataset, paired tests are commonly

which are specifically tailored to the chosen splits. Such and related effects, where we "overoptimize" the validation performance without effective reward in improved generalization performance have been sometimes dubbed "overtuning" or "oversearching". For a more detailed discussion of this topic, including related work, see Section 5 and Appendix B. The practice of reshuffling resampling splits during HPO is generally neither discussed in the scientific literature nor HPO software tools.[2] To the best of our knowledge, only Lévesque (2018) investigated reshuffling train-validation splits for every new HPC. For both holdout and $M$-fold CV using reshuffled resampling splits resulted in, on average, slightly lower generalization error when used in combination with Bayesian optimization (BO, Garnett, 2023) or CMA-ES (Hansen & Ostermeier, 2001) as HPO algorithms. Additionally, reshuffling was used by a solution to the NeurIPS 2006 performance prediction challenge to estimate the final generalization performance (Guyon et al., 2006). Recently, in the context of evolutionary optimization, reshuffling was applied after every generation (Larcher & Barbosa, 2022).

In this paper, we systematically examine the effect of reshuffling on HPO performance. Our contributions can be summarized as follows:

1. We show theoretically that reshuffling resampling splits during HPO can result in finding a configuration with better overall generalization performance, especially when the loss surface is rather flat and its estimate is noisy (Section 2).

2. We confirm these theoretical insights through controlled simulation studies (Section 3).

3. We demonstrate in realistic HPO benchmark experiments that reshuffling splits can lead to a real-world improvement of HPO (Section 4). Especially in the case of reshuffled holdout, we find that the final generalization performance is often on par with 5-fold CV under a wide range of settings.

We discuss results, limitations, and avenues for future research in Section 5.

## 2 Theoretical Analysis

### 2.1 Problem Statement and Setup

Machine learning (ML) aims to fit a model to data, so that it generalizes well to new observations of the same distribution. Let $\mathcal{D} = \{\boldsymbol{Z}_i\}_{i=1}^{n}$ be the observed dataset consisting of *i.i.d.* random variables from a distribution $P$, i.e., in the supervised setting $\boldsymbol{Z}_i = (\boldsymbol{X}_i, Y_i)$.[3,4] Formally, an inducer $g$ configured by an HPC $\boldsymbol{\lambda} \in \Lambda$ maps a dataset $\mathcal{D}$ to a model from our hypothesis space $h = g_{\boldsymbol{\lambda}}(\mathcal{D}) \in \mathcal{H}$. During HPO, we want to find a HPC that minimizes the expected generalization error, i.e., find

$$\boldsymbol{\lambda}^* = \arg\min_{\boldsymbol{\lambda} \in \Lambda} \mu(\boldsymbol{\lambda}), \quad \text{where} \quad \mu(\boldsymbol{\lambda}) = \mathbb{E}[\ell(\boldsymbol{Z}, g_{\boldsymbol{\lambda}}(\mathcal{D}))],$$

where $\ell(\boldsymbol{Z}, h)$ is the loss of model $h$ on a fresh observation $\boldsymbol{Z}$. In practice, there is usually a limited computational budget for each HPO run, so we assume that there is only a finite number of distinct HPCs $\Lambda = \{\boldsymbol{\lambda}_1, \dots, \boldsymbol{\lambda}_J\}$ to be evaluated, which also simplifies the subsequent analysis. Naturally, we cannot optimize the generalization error directly, but only an estimate of it. To do so, a resampling is constructed. For every HPC $\boldsymbol{\lambda}_j$, draw $M$ random sets $\mathcal{I}_{1,j}, \dots, \mathcal{I}_{M,j} \subset \{1, \dots, n\}$ of validation indices with $n_{\text{valid}} = \lceil \alpha n \rceil$ instances each. The random index draws are assumed to be independent of the observed data. The data is then split accordingly into pairs $\mathcal{V}_{m,j} = \{\boldsymbol{Z}_i\}_{i \in \mathcal{I}_{m,j}}, \mathcal{T}_{m,j} = \{\boldsymbol{Z}_i\}_{i \notin \mathcal{I}_{m,j}}$ of disjoint validation and training sets. Define the validation loss on the $m$-th fold

$$L(\mathcal{V}_{m,j}, g_{\boldsymbol{\lambda}_j}(\mathcal{T}_{m,j})) = \frac{1}{n_{\text{valid}}} \sum_{i \in \mathcal{I}_{m,j}} \ell(\boldsymbol{Z}_i, g_{\boldsymbol{\lambda}_j}(\mathcal{T}_{m,j})),$$

---

employed that implicitly assume that differences between the performance of classifiers on a given CV fold are comparable (Dietterich, 1998; Nadeau & Bengio, 1999, 2003; Demšar, 2006).

[2]In Appendix B, we present an overview of how resampling is addressed in tutorials and examples of standard HPO libraries and software. We conclude that usually fixed splits are used or recommended.

[3]Throughout, we use bold letters to indicate (fixed and random) vectors.

[4]We provide a notation table for symbols used in the main paper in Table 2 in the appendix.

and the $M$-fold validation loss as

$$\widehat{\mu}(\boldsymbol{\lambda}_j) = \frac{1}{M} \sum_{m=1}^{M} L(\mathcal{V}_{m,j}, g_{\boldsymbol{\lambda}_j}(\mathcal{T}_{m,j})).$$

Since $\mu$ is unknown, we minimize $\widehat{\boldsymbol{\lambda}} = \arg\min_{\boldsymbol{\lambda} \in \Lambda} \widehat{\mu}(\boldsymbol{\lambda})$, hoping that $\mu(\widehat{\boldsymbol{\lambda}})$ will also be small. Typically, the same splits are used for every HPC, so $\mathcal{I}_{m,j} = \mathcal{I}_m$ for all $j = 1, \ldots, J$ and $m = 1, \ldots, M$. In the following, we investigate how reshuffling train-validation splits (i.e., $\mathcal{I}_{m,j} \neq \mathcal{I}_{m,j'}$ for $j \neq j'$) affects the HPO problem.

## 2.2   How Reshuffling Affects the Loss Surface

We first investigate how different validation and reshuffling strategies affect the empirical loss surface $\widehat{\mu}$. In particular, we derive the limiting distribution of the sequence $\sqrt{n}(\widehat{\mu}(\boldsymbol{\lambda}_j) - \mu(\boldsymbol{\lambda}_j))_{j=1}^{J}$. This limiting regime will not only reveal the effect of reshuffling on the loss surface, but also give us a tractable setting to study HPO performance.

**Theorem 2.1.** *Under regularity conditions stated in Appendix C.1, it holds*

$$\sqrt{n}\left(\widehat{\mu}(\boldsymbol{\lambda}_j) - \mu(\boldsymbol{\lambda}_j)\right)_{j=1}^{J} \to \mathcal{N}(0, \Sigma) \quad \text{in distribution,}$$

*where*

$$\Sigma_{i,j} = \tau_{i,j,M} K(\boldsymbol{\lambda}_i, \boldsymbol{\lambda}_j), \quad \tau_{i,j,M} = \lim_{n \to \infty} \frac{1}{nM^2\alpha^2} \sum_{s=1}^{n} \sum_{m=1}^{M} \sum_{m'=1}^{M} \Pr(s \in \mathcal{I}_{m,i} \cap \mathcal{I}_{m',j}),$$

*and*

$$K(\boldsymbol{\lambda}_i, \boldsymbol{\lambda}_j) = \lim_{n \to \infty} \mathsf{Cov}[\bar{\ell}_n(\boldsymbol{Z}', \boldsymbol{\lambda}_i), \bar{\ell}_n(\boldsymbol{Z}', \boldsymbol{\lambda}_j)], \quad \bar{\ell}_n(\boldsymbol{z}, \boldsymbol{\lambda}) = \mathbb{E}[\ell(\boldsymbol{z}, g_{\boldsymbol{\lambda}}(\mathcal{T}))] - \mathbb{E}[\ell(\boldsymbol{Z}, g_{\boldsymbol{\lambda}}(\mathcal{T}))],$$

*where the expectation is taken over a training set $\mathcal{T}$ of size $n$ and two fresh samples $\boldsymbol{Z}, \boldsymbol{Z}'$ from the same distribution.*

The regularity conditions are rather mild and discussed further in Appendix C.1. The kernel $K$ reflects the (co-)variability of the losses caused by validation samples. The contribution of training samples only has a higher-order effect. The validation scheme enters the distribution through the quantities $\tau_{i,j,M}$. In what follows, we compute explicit expressions for some popular examples. The following list provides formal definitions for the index sets $\mathcal{I}_{m,j}$.

(i) (holdout) Let $M = 1$ and $\mathcal{I}_{1,j} = \mathcal{I}_1$ for all $j = 1, \ldots, J$, and some size-$\lceil \alpha n \rceil$ index set $\mathcal{I}_1$.

(ii) (reshuffled holdout) Let $M = 1$ and $\mathcal{I}_{1,1}, \ldots, \mathcal{I}_{1,J}$ be independently drawn from the uniform distribution over all size-$\lceil \alpha n \rceil$ subsets from $\{1, \ldots, n\}$.

(iii) ($M$-fold CV) Let $\alpha = 1/M$ and $\mathcal{I}_1, \ldots, \mathcal{I}_M$ be a disjoint partition of $\{1, \ldots, n\}$, and $\mathcal{I}_{m,j} = \mathcal{I}_m$ for all $j = 1, \ldots, J$.

(iv) (reshuffled $M$-fold CV) Let $\alpha = 1/M$ and $(\mathcal{I}_{1,j}, \ldots, \mathcal{I}_{M,j}), j = 1, \ldots, J$, be independently drawn from the uniform distribution over disjoint partitions of $\{1, \ldots, n\}$.

(v) ($M$-fold holdout) Let $\mathcal{I}_m, m = 1, \ldots, M$, be independently drawn from the uniform distribution over size-$\lceil \alpha n \rceil$ subsets of $\{1, \ldots, n\}$ and set $\mathcal{I}_{m,j} = \mathcal{I}_m$ for all $m = 1, \ldots, M, j = 1, \ldots, J$.

(vi) (reshuffled $M$-fold holdout) Let $\mathcal{I}_{m,j}, m = 1, \ldots, M, j = 1, \ldots, J$, be independently drawn from the uniform distribution over size-$\lceil \alpha n \rceil$ subsets of $\{1, \ldots, n\}$.

The value of $\tau_{i,j,M}$ for each example is computed explicitly in Appendix E. In all these examples, we in fact have

$$\tau_{i,j,M} = \begin{cases} \sigma^2, & i = j \\ \tau^2 \sigma^2, & i \neq j \end{cases}, \tag{1}$$

for some method-dependent parameters $\sigma, \tau$ shown in Table 1. The parameter $\sigma^2$ captures any increase in variance caused by omitting an observation from the validation sets. The parameter $\tau$ quantifies a potential decrease in correlation in the loss surface due to reshuffling. More precisely,

Table 1: Exemplary parametrizations in Equation (1) for resamplings; see Appendix E for details.

| Method | $\sigma^2$ | $\tau^2$ |
|---|---|---|
| holdout (HO) | $1/\alpha$ | $1$ |
| reshuffled HO | $1/\alpha$ | $\alpha$ |
| $M$-fold CV | $1$ | $1$ |
| reshuffled $M$-fold CV | $1$ | $1$ |
| $M$-fold HO (subsampling / Monte Carlo CV) | $1 + (1-\alpha)/M\alpha$ | $1$ |
| reshuffled $M$-fold HO | $1 + (1-\alpha)/M\alpha$ | $1/(1 + (1-\alpha)/M\alpha)$ |

the observed losses $\widehat{\mu}(\boldsymbol{\lambda}_i), \widehat{\mu}(\boldsymbol{\lambda}_j)$ at distinct HPCs $\boldsymbol{\lambda}_i \neq \boldsymbol{\lambda}_j$ become less correlated when $\tau$ is small. Generally, an increase in variance leads to worse generalization performance. The effect of a correlation decrease is less obvious and is studied in detail in the following section.

We make the following observations about the differences between methods in Table 1:

- $M$-fold CV incurs no increase in variance ($\sigma^2 = 1$) and — because every HPC uses the same folds — no decrease in correlation. Interestingly, the correlation does not even decrease when reshuffling the folds. In any case, all samples are used exactly once as validation and training instance. At least asymptotically, this leads to the same behavior, and reshuffling should have almost no effect on $M$-fold CV.

- The two (1-fold) holdout methods bear the same $1/\alpha$ increase in variance. This is caused by only using a fraction $\alpha$ of the data as validation samples. Reshuffled holdout also decreases the correlation parameter $\tau^2$. In fact, if HPCs $\boldsymbol{\lambda}_i \neq \boldsymbol{\lambda}_j$ are evaluated on largely distinct samples, the validation losses $\widehat{\mu}(\boldsymbol{\lambda}_i)$ and $\widehat{\mu}(\boldsymbol{\lambda}_j)$ become almost independent.

- $M$-fold holdout also increases the variance, because some samples may still be omitted from validation sets. This increase is much smaller for large $M$. Accordingly, the correlation is also decreased by less in the reshuffled variant.

## 2.3 How Reshuffling Affects HPO Performance

In practice, we are mainly interested in the performance of a model trained with the optimal HPC $\widehat{\boldsymbol{\lambda}}$. To simplify the analysis, we explore this in the large-sample regime derived in the previous section. Assume

$$\widehat{\mu}(\boldsymbol{\lambda}_j) = \mu(\boldsymbol{\lambda}_j) + \epsilon(\boldsymbol{\lambda}_j) \tag{2}$$

where $\epsilon(\boldsymbol{\lambda})$ is a zero-mean Gaussian process with covariance kernel

$$\mathsf{Cov}(\epsilon(\boldsymbol{\lambda}), \epsilon(\boldsymbol{\lambda}')) = \begin{cases} K(\boldsymbol{\lambda}, \boldsymbol{\lambda}) & \text{if } \boldsymbol{\lambda} = \boldsymbol{\lambda}', \\ \tau^2 K(\boldsymbol{\lambda}, \boldsymbol{\lambda}') & \text{else.} \end{cases} \tag{3}$$

Let $\Lambda \subseteq \{\boldsymbol{\lambda} \in \mathbb{R}^d \colon \|\boldsymbol{\lambda}\| \leq 1\}$ with $|\Lambda| = J < \infty$ be the set of hyperparameters. Theorem 2.2 ahead gives a bound on the expected regret $\mathbb{E}[\mu(\widehat{\boldsymbol{\lambda}}) - \mu(\boldsymbol{\lambda}^*)]$. It depends on several quantities characterizing the difficulty of the HPO problem. The constant

$$\kappa = \sup_{\|\boldsymbol{\lambda}\|, \|\boldsymbol{\lambda}'\| \leq 1} \frac{|K(\boldsymbol{\lambda}, \boldsymbol{\lambda}) - K(\boldsymbol{\lambda}, \boldsymbol{\lambda}')|}{K(\boldsymbol{\lambda}, \boldsymbol{\lambda})\|\boldsymbol{\lambda} - \boldsymbol{\lambda}'\|^2}.$$

can be interpreted as a measure of correlation of the process $\epsilon$. In particular, $\mathsf{Corr}(\epsilon(\boldsymbol{\lambda}), \epsilon(\boldsymbol{\lambda}')) \geq 1 - \kappa\|\boldsymbol{\lambda} - \boldsymbol{\lambda}'\|^2$. The constant is small when $\epsilon$ is strongly correlated, and large otherwise. Further, define $\eta$ as the minimal number such that any $\eta$-ball contained in $\{\|\boldsymbol{\lambda}\| \leq 1\}$ contains at least one element of $\Lambda$. It measures how densely the set of candidate HPCs $\Lambda$ covers set of all possible HPCs. If $\Lambda$ is a deterministic uniform grid, we have about $\eta \approx J^{-1/d}$. Similarly, Lemma D.1 in the Appendix shows that $\eta \lesssim J^{-1/2d}$ when randomly sampling HPCs. Finally, the constant

$$m = \sup_{\boldsymbol{\lambda} \in \Lambda} \frac{|\mu(\boldsymbol{\lambda}) - \mu(\boldsymbol{\lambda}^*)|}{\|\boldsymbol{\lambda} - \boldsymbol{\lambda}^*\|^2},$$

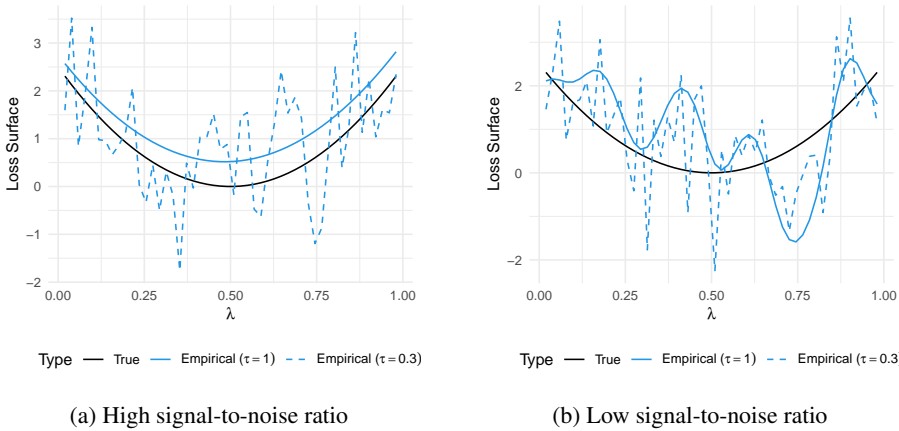

(a) High signal-to-noise ratio   (b) Low signal-to-noise ratio

Figure 1: Example of reshuffled empirical loss yielding a worse (left) and better (right) minimizer.

measures the local curvature at the minimum of the loss surface $\mu$. Finding an HPC $\boldsymbol{\lambda}$ close to the theoretical optimum $\boldsymbol{\lambda}^*$ is easier when the minimum is more pronounced (large $m$). On the other hand, the regret $\mu(\boldsymbol{\lambda}) - \mu(\boldsymbol{\lambda}^*)$ is also punishing mistakes more quickly. Defining $\log(x)_+ = \max\{0, \log(x)\}$, we can now state our main result.

**Theorem 2.2.** *Let $\widehat{\mu}$ follow the Gaussian process model* (2). *Suppose $\kappa < \infty$, $0 < \underline{\sigma}^2 \leq \mathsf{Var}[\epsilon(\boldsymbol{\lambda})] \leq \sigma^2 < \infty$ for all $\boldsymbol{\lambda} \in \Lambda$, and $m > 0$. Then*

$$\mathbb{E}[\mu(\widehat{\boldsymbol{\lambda}}) - \mu(\boldsymbol{\lambda}^*)] \leq \sigma\sqrt{d}[8 + B(\tau) - A(\tau)].$$

*where*

$$B(\tau) = 48\left[\sqrt{1-\tau^2}\sqrt{\log J} + \tau\sqrt{1 + \log(3\kappa)_+}\right], \quad A(\tau) = \sqrt{1-\tau^2}(\underline{\sigma}/\sigma)\sqrt{\log\left(\frac{\sigma}{2m\eta^2}\right)_+}.$$

The numeric constants result from several simplifications in a worst-case analysis, which lowers their practical relevance. A qualitative analysis of the bound is still insightful. The bound is increasing in $\sigma$ and $d$, indicating that the HPO problem is harder when there is a lot of noise or there are many parameters to tune. The terms $B(\tau)$ and $A(\tau)$ have conceptual interpretations:

- The term $B(\tau)$ quantifies how likely it is to pick a bad $\widehat{\boldsymbol{\lambda}}$ because of bad luck: a $\boldsymbol{\lambda}$ far away from $\boldsymbol{\lambda}^*$ had such a small $\epsilon(\boldsymbol{\lambda})$ that it outweighs the increase in $\mu$. Such events are more likely when the process $\epsilon$ is weakly correlated. Accordingly, $B(\tau)$ is decreasing in $\tau$ and increasing in $\kappa$.

- The term $A(\tau)$ quantifies how likely it is to pick a good $\widehat{\boldsymbol{\lambda}}$ by luck: a $\boldsymbol{\lambda}$ close to $\boldsymbol{\lambda}^*$ had such a small $\epsilon(\boldsymbol{\lambda})$ that it overshoots all the other fluctuations. Also such events are more likely when the process $\epsilon$ is weakly correlated. Accordingly, the term $A(\tau)$ is decreasing in $\tau$.

The $B$, as stated, is unbounded, but a closer inspection of the proof shows that it is upper bounded by $\sqrt{\log J}$. This bound is attained only in the unrealistic scenario when the validation losses are essentially uncorrelated across all HPCs. The term $A$ is bounded from below by zero, which is also the worst case because the term enters our regret bound with a negative sign.

Both $A$ and $B$ are decreasing in the reshuffling parameter $\tau$. There are two regimes. If $\sigma/2m\eta^2 \leq e$, then $A(\tau) = 0$ and reshuffling cannot lead to an improvement of the bound. The term $\sigma/m\eta^2$ can be interpreted as noise-to-signal ratio (relative to the grid density). If the signal is much stronger than the noise, the HPO problem is so easy that reshuffling will not help. This situation is illustrated in Figure 1a.

If on the other hand $\sigma/m\eta^2 > e$, the terms $A(\tau)$ and $B(\tau)$ enter the bound with opposing signs. This creates tension: reshuffling between HPCs increases $B(\tau)$, which is countered by a decrease in $A(\tau)$. So which scenarios favor reshuffling? When the process $\epsilon$ is strongly correlated, $\kappa$ is small and reshuffling (decreasing $\tau$) incurs a high cost in $B(\tau)$. This is intuitive: When there is strong

correlation, the validation loss surface $\widehat{\mu}$ is essentially just a vertical shift of $\mu$. Finding the optimal $\boldsymbol{\lambda}$ is then almost as easy as if we would know $\mu$, and decorrelating the surface through reshuffling would make it unnecessarily hard. When $\epsilon$ is less correlated ($\kappa$ large) however, reshuffling does not hurt the term $B(\tau)$ as much, but we can reap all the benefits of increasing $A(\tau)$. Here, the effect of reshuffling can be interpreted as hedging against the catastrophic case where all $\widehat{\mu}(\boldsymbol{\lambda})$ close to the optimal $\boldsymbol{\lambda}^*$ are simultaneously dominated by a region of bad hyperparameters. This is illustrated in Figure 1b.

# 3 Simulation Study

To test our theoretical understanding of the potential benefits of reshuffling resampling splits during HPO, we conduct a simulation study. This study helps us explore the effects of reshuffling in a controlled setting.

## 3.1 Design

We construct a univariate quadratic loss surface function $\mu : \Lambda \subset \mathbb{R} \mapsto \mathbb{R}, \lambda \to m(\lambda - 0.5)^2/2$ which we want to minimize. The global minimum is given at $\mu(0.5) = 0$. Combined with a kernel for the noise process $\epsilon$ as in Equation (3), this allows us to simulate an objective as observed during HPO by sampling $\widehat{\mu}(\lambda) = \mu(\lambda) + \epsilon(\lambda)$. We use a squared exponential kernel $K(\lambda, \lambda') = \sigma_K^2 \exp\left(-\kappa(\lambda - \lambda')^2/2\right)$ that is plugged into the covariance kernel of the noise process $\epsilon$ in Equation (3). The parameters $m$ and $\kappa$ in our simulation setup correspond exactly to the curvature and correlation constants from the previous sections. Recall that Theorem 2.2 states that the effect of reshuffling strongly depends on the curvature $m$ of the loss surface $\mu$ (a larger $m$ implies a stronger curvature) and the constant $\kappa$ as a measure of correlation of the noise $\epsilon$ (a larger $\kappa$ implies weaker correlation). Combined with the possibility to vary $\tau$ in the covariance kernel of $\epsilon$, we can systematically investigate how curvature of the loss surface, correlation of the noise and the extent of reshuffling affect optimization performance. In each simulation run, we simulate the observed objective $\hat{\mu}(\lambda)$, identify the minimizer $\hat{\lambda} = \arg\min_{\lambda \in \Lambda} \hat{\mu}(\lambda)$, and calculate its true risk, $\mu(\hat{\lambda})$. We repeat this process 10000 times for various combinations of $\tau$, $m$, and $\kappa$.

## 3.2 Results

Figure 2 visualizes the true risk of the configuration $\hat{\lambda}$ that minimizes the observed objective. We observe that for a loss surface with low curvature (i.e., $m \leq 2$), reshuffling is beneficial (lower values of $\tau$ resulting in a better true risk of the configuration that optimizes the observed objective) as long as the noise process is not too correlated (i.e., $\kappa \geq 1$). As soon as the noise process is more strongly correlated, even flat valleys of the true risk $\mu$ remain clearly visible in the observed risk $\widehat{\mu}$, and reshuffling starts to hurt the optimization performance. Moving to scenarios of high curvature, the general relationship of $m$ and $\kappa$ remains the same, but reshuffling starts to hurt optimization performance already with weaker correlation in the noise. In summary, the simulations show that in cases of low curvature of the loss surface, reshuffling (reducing $\tau$) tends to improve the true risk of the optimized configuration, especially when the loss surface is flat (small $m$) and the noise is not strongly correlated (i.e., $\kappa$ is large). This exactly confirms our theoretical predictions from the previous section.

# 4 Benchmark Experiments

In this section, we present benchmark experiments of real-world HPO problems where we investigate the effect of reshuffling resampling splits during HPO. First, we discuss the experimental setup. Second, we present results for HPO using random search (Bergstra & Bengio, 2012). Third, we also show the effect of reshuffling when applied in BO using HEBO (Cowen-Rivers et al., 2022) and SMAC3 (Lindauer et al., 2022). Recall that our theoretical insight suggests that 1) reshuffling might be beneficial during HPO and 2) holdout should be affected the most by reshuffling and other resamplings should only be affected to a lesser extent.

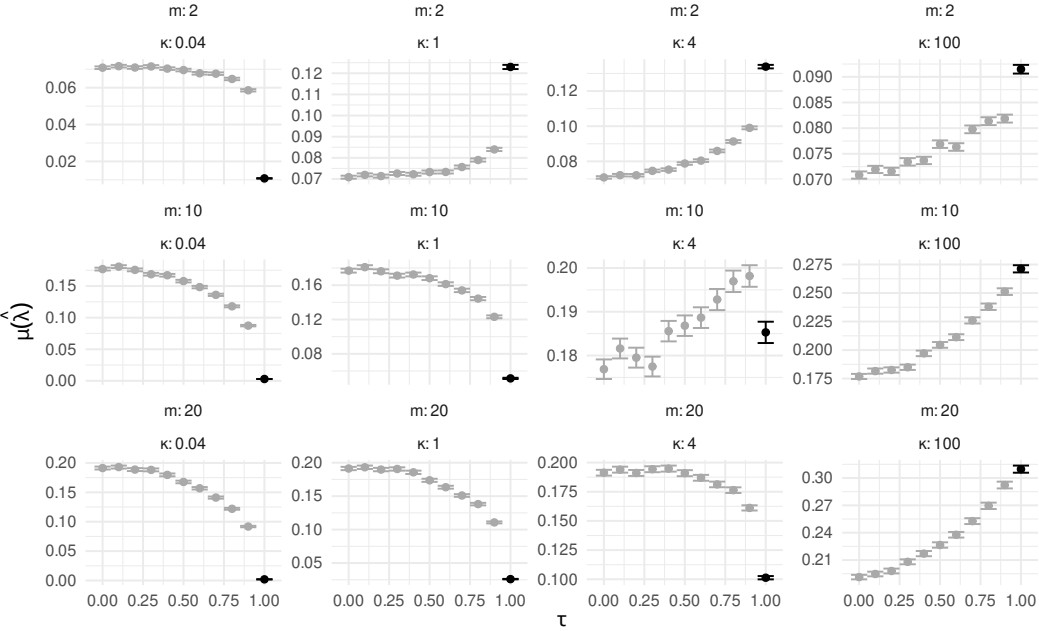

Figure 2: Mean true risk (lower is better) of the configuration minimizing the observed objective systematically varied with respect to curvature $m$, correlation strength $\kappa$ of the noise (a larger $\kappa$ implying weaker correlation), and extent of reshuffling $\tau$ (lower $\tau$ increasing reshuffling). A $\tau$ of 1 indicates no reshuffling. Error bars represent standard errors.

## 4.1 Experimental Setup

As benchmark tasks, we use a set of standard HPO problems defined on small- to medium-sized tabular datasets for binary classification. We suspect the effect of the resampling variant used and whether the resampling is reshuffled to be larger for smaller datasets, where the variance of the validation loss estimator is naturally higher. Furthermore, from a practical perspective, this also ensures computational feasibility given the large number of HPO runs in our experiments. We systematically vary the learning algorithm, optimized performance metric, resampling method, whether the resampling is reshuffled, and the size of the dataset used for training and validation during HPO. Below, we outline the general experimental design and refer to Appendix F for details.

We used a subset of the datasets defined by the AutoML benchmark (Gijsbers et al., 2024), treating these as data generating processes (DGPs; Hothorn et al., 2005). We only considered datasets with less than 100 features to reduce the required computation time and required the number of observations to be between 10000 and 1000000; for further details see Appendix F.1. Our aim was to robustly measure the generalization performance when varying the size $n$, which, as defined in Section 2 denotes the size of the combined data for model selection, so one training and validation set combined. First, we sampled 5000 data points per dataset for robust assessment of the generalization error; these points are not used during HPO in any way. Then, from the remaining points we sampled tasks with $n \in \{500, 1000, 5000\}$.

We selected CatBoost (Prokhorenkova et al., 2018) and XGBoost (Chen & Guestrin, 2016) for their state-of-the-art performance on tabular data (Grinsztajn et al., 2022; Borisov et al., 2022; McElfresh et al., 2023; Kohli et al., 2024). Additionally, we included an Elastic Net (Zou & Hastie, 2005) to represent a linear baseline with a smaller search space and a funnel-shaped MLP (Zimmer et al., 2021) as a cost-effective neural network baseline. We provide details regarding training pipelines and search spaces in Appendix F.2.

We conduct a random search with 500 HPC evaluations for every resampling strategy we described in Table 1, for both fixed and reshuffled splits. We always use 80/20 train-validation splits for holdout

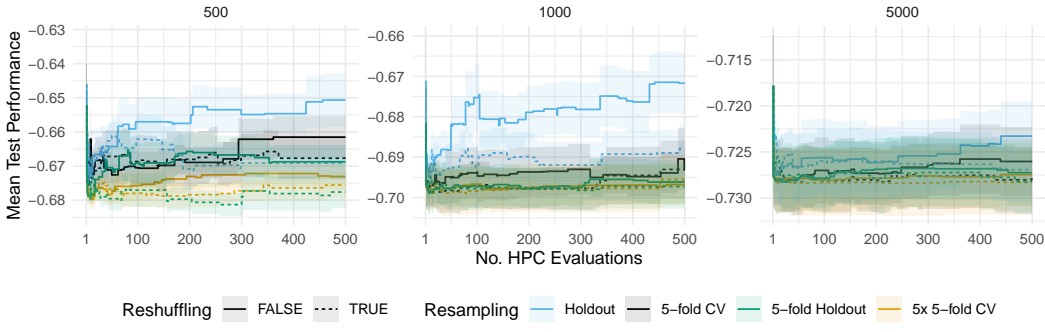

Figure 3: Average test performance (negative ROC AUC) of the incumbent for XGBoost on dataset albert for increasing $n$ (train-validation sizes, columns). Shaded areas represent standard errors.

and 5-fold CVs, so that training set size (and negative estimation bias) are the same. Anytime test performance of an HPO run is assessed by re-training the current incumbent (i.e. the best HPC until the current HPO iteration based on validation performance) on all available train and validation data and evaluating its performance on the outer test set. Note we do this for scientific evaluation in this experiment; obviously, this is not possible in practice. Using random search allows us to record various metrics and afterwards simulate optimizing for different ones, specifically, we recorded accuracy, area under the ROC curve (ROC AUC) and logloss.

We also investigated the effect of reshuffling on two state-of-the-art BO variants (Eggensperger et al., 2021; Turner et al., 2021), namely HEBO (Cowen-Rivers et al., 2022) and SMAC3 (Lindauer et al., 2022). The experimental design was the same as for random search, except for the budget, which we reduced from 500 HPCs to 250 HPCs, and only optimized ROC AUC.

## 4.2 Experimental Results

In the following, we focus on the results obtained using ROC AUC. We present aggregated results over different tasks, learning algorithms and replications to get a general understanding of the effects. Unaggregated results and results involving accuracy and logloss can be found in Appendix G.

**Results of Reshuffling Different Resamplings** For each resampling (holdout, 5-fold holdout, 5-fold CV, and 5x 5-fold CV), we empirically analyze the effect of reshuffling train and validation splits during HPO.

In Figure 3 we exemplarily show how test performance develops over the course of an HPO run on a single task for different resamplings (with and without reshuffling). Naturally, test performance does not necessarily increase in a monotonic fashion, and especially holdout without reshuffling tends to be unstable. Its reshuffled version results in substantially better test performance.

Next, we look at the relative *improvement* (compared to standard 5-fold CV, which we consider our baseline) with respect to *test* ROC AUC performance of the incumbent over time in Figure 4, i.e., the difference in test performance of the incumbent between standard 5-fold CV and a different resampling protocol; hence a positive difference tells us how much better in test error we are, if we would have chosen the other protocol instead 5-fold CV. We observe that reshuffling generally results in equal or better performance compared to the same resampling protocol without reshuffling. For 5-fold holdout and especially 5-fold CV and 5x 5-fold CV, reshuffling has a smaller effect on relative test performance improvement, as expected. Holdout is affected the most by reshuffling and results in substantially better relative test performance compared to standard holdout. We also observe that an HPO protocol based on reshuffled holdout results in similar final test performance as standard 5-fold CV while overall being substantially cheaper due to requiring less model fits per HPC evaluation. In Appendix G.2, we further provide an ablation study on the number of folds when using $M$-fold holdout, where we observed that – in line with our theory – the more folds are used, the less reshuffling affects $M$-fold holdout.

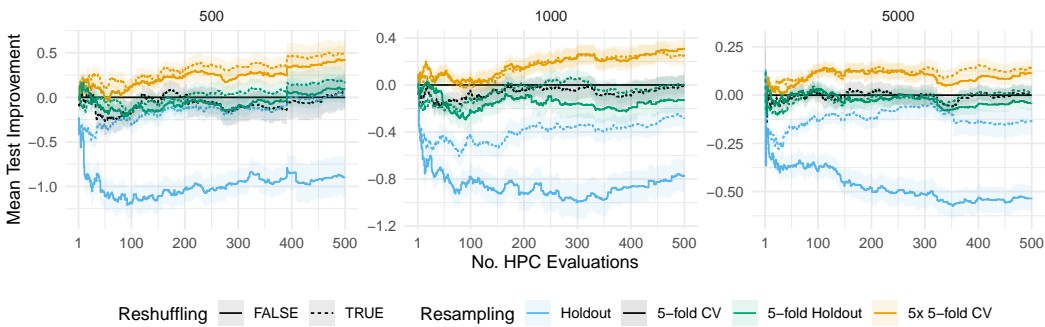

Figure 4: Average improvement (compared to standard 5-fold CV) with respect to test performance (ROC AUC) of the incumbent over different tasks, learning algorithms and replications separately for increasing $n$ (train-validation sizes, columns). Shaded areas represent standard errors.

However, this general trend can vary for certain combinations of classifier and performance metric, see Appendix G. Especially for logloss, we observed that reshuffling rarely is beneficial; see the discussion in Section 5. Finally, the different resamplings generally behave as expected. The more we are willing to invest compute resources into a more intensive resampling like 5-fold CV or 5x 5-fold CV, the better the generalization performance of the final incumbent.

**Results for BO and Reshuffling** Figure 5 shows that, generally HEBO and SMAC3 outperform random search with respect to generalization performance (i.e., comparing HEBO and SMAC3 to random search under standard holdout, or comparing under reshuffled holdout). More interestingly, HEBO, SMAC3 and random search all strongly benefit from reshuffling. Moreover, the performance gap between HEBO and random search but also SMAC3 and random search narrows when the resampling is reshuffled, which is an interesting finding of its own: As soon as we are concerned with generalization performance of HPO and not only investigate validation performance during optimization, the choice of optimizer might have less impact on final generalization performance compared to other choices such as whether the resampling is reshuffled during HPO or not. We present results for BO and reshuffling for different resamplings in Appendix G.

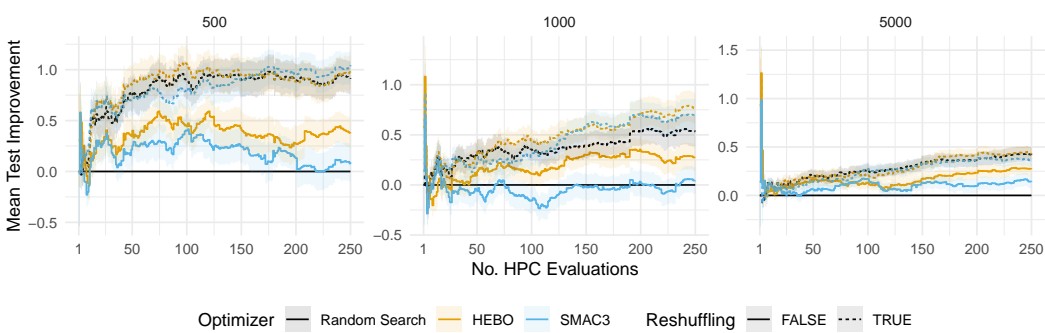

Figure 5: Average improvement (compared to random search on standard holdout) with respect to test performance (ROC AUC) of the incumbent over tasks, learning algorithms and replications for different $n$ (train-validation sizes, columns). Shaded areas represent standard errors.

## 5 Discussion

In the previous sections, we have shown theoretically and empirically that reshuffling can enhance generalization performance of HPO. The main purpose of this article is to draw attention to this

surprising fact about a technique that is simple but rarely discussed. Our work goes beyond a preliminary experimental study on reshuffling (Lévesque, 2018), in that we also study the effect of reshuffling on random search, multiple metrics and learning algorithms, and most importantly, for the first time, we provide a theoretical analysis that explains why reshuffling can be beneficial.

**Limitations**  To unveil the mechanisms underlying the reshuffling procedures, our theoretical analysis relies on an asymptotic approximation of the empirical loss surface. This allows us to operate on Gaussian loss surfaces, which exhibit convenient concentration and anti-concentration properties required in our proof. The latter are lacking for general distributions, which explains our asymptotic approach. The analysis was further facilitated by a loss stability assumption regarding the learning algorithms that is generally rather mild; see the discussion in Bayle et al. (2020). However, it typically fails for highly sensitive losses, which has practical consequences. In fact, Figure 9 in Appendix G shows that reshuffling usually hurts generalization for the logloss and small sample sizes. It is still an open question whether this problem can be fixed by less naive implementations of the technique. Another limitation is our focus on generalization after search through a fixed, finite set of candidates. This largely ignores the dynamic nature of many HPO algorithms, which would greatly complicate our analysis. Finally, our experiments are limited in that we restricted ourselves to tabular data and binary classification and we avoided extremely small or large datasets.

**Relation to Overfitting**  The fact that generalization performance can decrease during HPO (or computational model selection in general) is sometimes known as oversearching, overtuning, or overfitting to the validation set (Quinlan & Cameron-Jones, 1995; Escalante et al., 2009; Koch et al., 2010; Igel, 2012; Bischl et al., 2023), but has arguably not been studied very thoroughly. Given recent theoretical (Feldman et al., 2019) and empirical (Purucker & Beel, 2023) findings, we expect less overtuning on multi-class datasets, making it interesting to see how reshuffling would affect the generalization performance.

Several works suggest strategies to counteract this effect. First, LOOCVCV proposes a conservative choice of incumbents (Ng, 1997) at the cost of leave-one-out analysis or an additional hyperparameter. Second, it is possible to use an extra *selection set* (Igel, 2012; Lévesque, 2018; Mohr et al., 2018) at the cost of reduced training data, which was found to lead to reduced overall performance (Lévesque, 2018). Third, by using early stopping one can stop hyperparameter optimization before the generalization performance degrades again. This was so far demonstrated to be able to save compute budget at only marginally reduced performance, but also requires either a sensitivity hyperparameter or correct estimation of the variance of the generalization estimate and was only developed for cross-validation so far (Makarova et al., 2022). Reshuffling itself is orthogonal to these proposals and a combination with the above-mentioned methods might result in further improvements.

**Outlook**  Generally, the related literature detects overfitting to the validation set either visually (Ng, 1997) or by measuring it (Koch et al., 2010; Igel, 2012; Fabris & Freitas, 2019). Developing a unified formal definition of the above-mentioned terms and thoroughly analyzing the effect of decreased generalization performance after many HPO iterations and how it relates to our measurements of the validation performance is an important direction for future work.

We further found, both theoretically and experimentally, that investing more resources when evaluating each HPC can result in better final HPO performance. To reduce the computational burden on HPO again, we suggest further investigating the use of adaptive CV techniques, as proposed by Auto-WEKA (Thornton et al., 2013) or under the name Lazy Paired Hyperparameter Tuning (Zheng & Bilenko, 2013). Designing more advanced HPO algorithms exploiting the reshuffling effect should be a promising avenue for further research.

## Acknowledgments and Disclosure of Funding

We thank Martin Binder and Florian Karl for helpful discussions. Lennart Schneider is supported by the Bavarian Ministry of Economic Affairs, Regional Development and Energy through the Center for Analytics - Data - Applications (ADACenter) within the framework of BAYERN DIGITAL II (20-3410-2-9-8). Lennart Schneider acknowledges funding from the LMU Mentoring Program of the Faculty of Mathematics, Informatics and Statistics.

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

# A Notation

Table 2: Notation table. We discuss all symbols used in the main paper.

| | |
|---|---|
| $\boldsymbol{X}_i$ | Random vector, describing the features |
| $Y_i$ | Random variable, describing the target |
| $\boldsymbol{Z}_i = (\boldsymbol{X}_i, Y_i)$ | Data point |
| $\mathcal{D} = \{\boldsymbol{Z}_i\}_{i=1}^n$ | Dataset consisting of *iid* random variables |
| $n$ | Number of observations |
| $g$ | Inducer/ML algorithm |
| $h$ | Model, created by the inducer via $h = g_{\boldsymbol{\lambda}}(\mathcal{D})$ |
| $\boldsymbol{\lambda}$ | Hyperparameter configuration |
| $\Lambda$ | Finite set of all hyperparameter configurations |
| $J$ | $|\Lambda|$, i.e., the number of hyperparameter configurations |
| $g_{\boldsymbol{\lambda}_j}$ | Hyperparameterized inducer |
| $\mu(\boldsymbol{\lambda})$ | Expected loss of a hyperparameterized inducer on the distribution of a dataset |
| $\ell(\boldsymbol{Z}, h)$ | Loss of a model $h$ on a fresh observation $\boldsymbol{Z}$ |
| $M$ | Number of folds in M-fold cross-validation |
| $\alpha$ | Percentage of samples to be used for validation |
| $\mathcal{I}_{1,j}, \ldots, \mathcal{I}_{M,j} \subset \{1, \ldots, n\}$ | $M$ sets of validation indices, to be used for evaluating $\boldsymbol{\lambda}_j$ |
| $\mathcal{V}_{m,j}$ | Validation data for fold $m$ and configuration $\boldsymbol{\lambda}_j$ |
| $\mathcal{T}_{m,j}$ | Training data for fold $m$ and configuration $\boldsymbol{\lambda}_j$ |
| $L(\mathcal{V}_{m,j}, g_{\boldsymbol{\lambda}_j}(\mathcal{T}_{m,j}))$ | Validation loss for fold $m$ and configuration $\boldsymbol{\lambda}_j$ |
| $\widehat{\mu}(\boldsymbol{\lambda}_j)$ | M-fold validation loss |
| $\sigma^2$ | Increase in variance of validation loss caused by resampling |
| $\tau^2$ | Decrease in correlation among validation losses caused by reshuffling |
| $\tau_{i,j,M}$ | Resampling-related component of validation loss covariance |
| $K(\cdot, \cdot)$ | Kernel capturing the covariance of the pointwise losses between two HPCs |
| $\epsilon(\boldsymbol{\lambda}_j)$ | Zero-mean Gaussian process, see Equation (2) |
| $d$ | Number of hyperparameters |
| $\kappa$ | Curvature constant of covariance kernel |
| $\eta$ | Density of hyperparameter set $\Lambda$ |
| $m$ | Local curvature at the minimum of the loss surface $\mu$ |
| $\underline{\sigma}$ | Lower bound on the noise level |
| $B(\tau)$ | Part of the regret bound penalizing reshuffling |
| $A(\tau)$ | Part of the regret bound rewarding reshuffling |

# B Extended Related Work

Due to the black box nature of the HPO problem (Feurer & Hutter, 2019; Bischl et al., 2023), gradient free, zeroth-order optimization algorithms such as BO (Garnett, 2023), Evolutionary Strategies (Loshchilov & Hutter, 2016) or a simple random search (Bergstra & Bengio, 2012) have become standard optimization algorithms to tackle vanilla HPO problems.

In the last decade, most research on HPO has been concerned with constructing new algorithms that excel at finding configurations with a low estimated generalization error. Examples include BO variants such as as HEBO (Cowen-Rivers et al., 2022) or SMAC3 (Lindauer et al., 2022). Another direction of HPO research has been concerned with speeding up the HPO process to allow more efficient spending of compute resources. Multifidelity HPO, for example, turns the black box optimization problem into a gray box one by making use of lower fidelity approximations to the target function, i.e., using fewer numbers of epochs or subsets of the data for cheap low-fidelity evaluations that approximate the costly high-fidelity evaluation. Examples include bandit-based budget allocation algorithms such as Successive Halving (Jamieson & Talwalkar, 2016), Hyperband (Li et al., 2018) and their extensions that use non-random search mechanisms (Falkner et al., 2018; Awad et al., 2021; Mallik et al., 2023) or algorithms making use of multi-fidelity information in the context of BO (Swersky et al., 2014; Klein et al., 2017; Wu et al., 2020; Kadra et al., 2023). Several works address the problem of speeding up cross-validation techniques and use techniques that could be described as grey box optimization techniques. Besides the ones mentioned in the main paper (Thornton et al., 2013; Zheng & Bilenko, 2013), it is possible to employ racing techniques for model selection in machine learning as demonstrated by Lang et al. (2015), and there has been a recent interest in methods that adapt the cost of running full cross-validation procedures (Bergman et al., 2024; Buczak et al., 2024).

When addressing the problem of HPO, we must acknowledge an inherent mismatch between the explicit objective we optimize – namely, the estimated generalization performance of a model – and the actual implicit optimization goal, which is to identify a configuration that yields the best

generalization performance on new, unseen data. Typically, evaluations and comparisons of different HPO algorithms focus exclusively on the final best validation performance (i.e., the objective that is directly optimized), even though an unbiased estimate of performance on an external unseen test set might be available. While this approach is logical for assessing the efficacy of an optimization algorithm based on the metric it seeks to improve, relying solely on finding an optimal validation configuration is beneficial only if there is reason to assume a strong correlation between the optimized validation performance and true generalization ability on new, unseen test data. This discrepancy can be found deeply within the HPO community, where the evaluation of HPO algorithms on standard benchmark libraries is usually done solely with respect to the validation performance (Eggensperger et al., 2021; Pineda Arango et al., 2021; Salinas et al., 2022; Pfisterer et al., 2022).[5] This relationship between validation performance (i.e., the estimated generalization error derived from resampling) and true generalization performance (e.g., assessed through an outer holdout test set or additional resampling) of an optimal validation configuration found during HPO remains a largely unexplored area of research.

In general, little research has focused on the selection of resampling types, let alone the automated selection of resampling types (Guyon et al., 2010; Feurer et al., 2022). While we usually expect that a more intensive resampling will reduce the variance of the estimated generalization error and thereby improve the (rank) correlation between optimized validation and unbiased outer test performance within HPCs, this benefit is naturally offset by a higher computational expense. Overall, there is little research on which resampling method to use in practice for model selection, and we only know of a study for support vector machines (Wainer & Cawley, 2017), a simulation study for clinical prediction models (Dunias et al., 2024), a study on feature selection (Molinaro et al., 2005) and a study on fast CV (Bergman et al., 2024). In addition, ML-Plan (Mohr et al., 2018) proposed a two-stage procedure. In a first stage (search), the tool uses planning on hierarchical task networks to find promising machine learning pipelines on $70\%$ of the training data. In a second step (selection), it uses $100\%$ of the training data and retrains the most promising candidates from the search step. Finally, it uses a combination of the internal generalization error estimation that was used during search and the $0.75$ percentile of the generalization error estimation from the selection step to make a more unbiased selection of the final model. The paper found that this improves performance over using only regular cross-validation for search and selection. The general consensus, that is in agreement with our findings, is that CV or repeated CV generally leads to better generalization performance. In addition, while there are theoretical works that compare the accuracy of estimating the generalization error of holdout and CV (Blum et al., 1999), our goals is to correctly identify a single solution, which generalizes well, see the excellent survey by Arlot & Celisse (2010) for a discussion on this topic.

Bouthillier et al. (2021) studied the sources of variance in machine learning experiments, and find that the split into training and test data has the largest impact. Consequently, they suggest to reshuffle the data prior to splitting it into the training, which is then used for HPO, and the test set. We followed their suggestion when designing our experiments and draw a new test sample for every replication, see Section 4.1 and Appendix F. This dependence on the exact split was further already discussed in the context of how much the outcome of a statistical test on results of machine learning experiments depended on the exact train-test split (Bouckaert, 2004).

Finally, the first warning against comparing too many hypothesis using cross-validation was raised by Schaffer (1993), and in addition to the works discussed in Section 5 in the main paper, also picked up by Rao et al. (2008); Cawley & Talbot (2010). Moreover, the problem of finding a correct "upper objective" in a bilevel optimization problem has been noted (Guyon et al., 2010, 2015, 2019). Also, in the related field of algorithm configuration the problem has been identified (Eggensperger et al., 2019).

### B.1 Current Treatment of Resamplings in HPO Libraries and Software

In Table 3, we provide a brief summary of how resampling is handled in popular HPO libraries and software.[6] For each library, we checked whether the core functionality, examples, or tutorials mention

---

[5]We admit that these benchmark libraries implement efficient benchmarking methods such as surrogate (Eggensperger et al., 2018; Pfisterer et al., 2022) or tabular benchmarks (Pineda Arango et al., 2021). It would be possible to adapt them to return the test performance, however, changes in the HPO evaluation protocol, such as the one we propose, would not be feasible.

[6]This summary is not exhaustive but reflects the general consensus observed in widely-used software.

the possibility of reshuffling the resampling during HPO or if the resampling is considered fixed. If reshuffling is used in an example, mentioned, or if core functionality uses it, we mark it with a ✓. If it is unclear or inconsistent across examples and core functionality, we mark it with a ?. Otherwise, we use a ✗. Our conclusion is that the concept of reshuffling resampling generally receives little attention.

Table 3: Exemplary Treatment of Resamplings in HPO Libraries and Software

| Software | Reshuffled? | Reference(s) |
|---|---|---|
| sklearn | ✗ | `GridSearchCV`[1]/ `RandomizedSearchCV`[2] |
| HEBO | ✗ | `sklearn_tuner`[3] |
| optuna | ? | Inconsistency between examples[4,5,6] |
| bayesian-optimization | ✗ | sklearn Example[7,8] |
| ax | ✗ | CNN Example[9] |
| spearmint | ✗ | No official HPO Examples |
| scikit-optimize | ✗ | BO for GBT Example[7,10] |
| SMAC3 | ✗ | SVM Example[7,11] |
| dragonfly | ✗ | Tree Based Ensemble Example[12] |
| aws sagemaker | ✗ | Blog Post[13] |
| raytune | ? | Inconsistency between examples[14,15] |
| hyperopt(-sklearn) | ? | Cost Function Logic[16] |

✗: no reshuffling, ?: both reshuffling and no reshuffling or unclear, ✓: reshuffling

[1] https://github.com/scikit-learn/scikit-learn/blob/8721245511de2f225ff5f9aa5f5fadce663cd4a3/sklearn/model_selection/_search.py#L1263

[2] https://github.com/scikit-learn/scikit-learn/blob/8721245511de2f225ff5f9aa5f5fadce663cd4a3/sklearn/model_selection/_search.py#L1644

[3] https://github.com/huawei-noah/HEBO/blob/b60f41aa862b4c5148e31ab4981890da6d41f2b1/HEBO/hebo/sklearn_tuner.py#L73

[4] https://github.com/optuna/optuna-integration/blob/15e6b0ec6d9a0d7f572ad387be8478c56257bef7/optuna_integration/sklearn/sklearn.py#L223 here sklearn's `cross_validate` is used which by default does not reshuffle the resampling https://github.com/scikit-learn/scikit-learn/blob/8721245511de2f225ff5f9aa5f5fadce663cd4a3/sklearn/model_selection/_validation.py#L186

[5] https://github.com/optuna/optuna-examples/blob/dd56b9692e6d1f4fa839332edbcdd93fd48c16d8/pytorch/pytorch_simple.py#L79 here, data loaders for train and valid are instantiated within the objective of the trial but the data within the loaders is fixed

[6] https://github.com/optuna/optuna-examples/blob/dd56b9692e6d1f4fa839332edbcdd93fd48c16d8/xgboost/xgboost_simple.py#L22 here, the train validation split is performed within the objective of the trial and no seed is set which results in reshuffling https://github.com/scikit-learn/scikit-learn/blob/8721245511de2f225ff5f9aa5f5fadce663cd4a3/sklearn/model_selection/_split.py#L2597

[7] functionality relies on sklearn's `cross_val_score` which by default does not reshuffle the resampling https://github.com/scikit-learn/scikit-learn/blob/8721245511de2f225ff5f9aa5f5fadce663cd4a3/sklearn/model_selection/_validation.py#L631

[8] https://github.com/bayesian-optimization/BayesianOptimization/blob/c7e5c3926944fc6011ae7ace29f7b5ed0f9c983b/examples/sklearn_example.py#L32

[9] https://github.com/facebook/Ax/blob/ac44a6661f535dd3046954f8fd8701327f4a53e2/tutorials/tune_cnn_service.ipynb#L39 and https://github.com/facebook/Ax/blob/ac44a6661f535dd3046954f8fd8701327f4a53e2/ax/utils/tutorials/cnn_utils.py#L154

[10] https://github.com/scikit-optimize/scikit-optimize/blob/a2369ddbc332d16d8ff173b12404b03fea472492/examples/hyperparameter-optimization.py#L82C21-L82C36

[11] https://github.com/automl/SMAC3/blob/9aaa8e94a5b3a9657737a87b903ee96c683cc42c/examples/1_basics/2_svm_cv.py#L63

[12] https://github.com/dragonfly/dragonfly/blob/3eef7d30bcc2e56f2221a624bd8ec7f933f81e40/examples/tree_reg/skltree.py#L111

[13] https://aws.amazon.com/blogs/architecture/field-notes-build-a-cross-validation-machine-learning-model-pipeline-at-scale-with-amazon-sagemaker/

[14] https://github.com/ray-project/ray/blob/3f5aa5c4642eeb12447d9de5dce22085512312f3/doc/source/tune/examples/tune-pytorch-cifar.ipynb#L120 here, data loaders for train and valid are instantiated within the objective but the data within the loaders are fixed

[15] https://github.com/ray-project/ray/blob/3f5aa5c4642eeb12447d9de5dce22085512312f3/doc/source/tune/examples/tune-xgboost.ipynb#L335 here, the train validation split is performed within the objective and no seed is set which results in reshuffling https://github.com/scikit-learn/scikit-learn/blob/8721245511de2f225ff5f9aa5f5fadce663cd4a3/sklearn/model_selection/_split.py#L2597

[16] https://github.com/hyperopt/hyperopt-sklearn/blob/4bc286479677a0bfd2178dac4546ea268b3f3b77/hpsklearn/estimator/_cost_fn.py#L144 dependence on random seed which by default is not set and there is no discussion of reshuffling and behavior is somewhat unclear

# C   Proofs of the Main Results

## C.1   Proof of Theorem 2.1

We impose *stability* assumptions on the learning algorithm similar to Bayle et al. (2020); Austern & Zhou (2020). Let $\boldsymbol{Z}, \boldsymbol{Z}_1, \ldots, \boldsymbol{Z}_n, \boldsymbol{Z}_1'$, be *iid* random variables. Define $\mathcal{T} = \{\boldsymbol{Z}_i\}_{i=1}^n$, and $\mathcal{T}'$ as $\mathcal{T}$ but with $\boldsymbol{Z}_n$ replaced by the independent copy $\boldsymbol{Z}_n'$. Define

$$\widetilde{\ell}_n(\boldsymbol{z}, \boldsymbol{\lambda}) = \ell(\boldsymbol{z}, g_{\boldsymbol{\lambda}}(\mathcal{T})) - \mathbb{E}[\ell(\boldsymbol{Z}, g_{\boldsymbol{\lambda}}(\mathcal{T})) \mid \mathcal{T}],$$

assume that each $g_{\boldsymbol{\lambda}}(\mathcal{T})$ is invariant to the ordering in $\mathcal{T}$, $\ell$ is bounded, and

$$\max_{\boldsymbol{\lambda} \in \Lambda} \mathbb{E}\{[\widetilde{\ell}(\boldsymbol{Z}, g_{\boldsymbol{\lambda}}(\mathcal{T})) - \widetilde{\ell}(\boldsymbol{Z}, g_{\boldsymbol{\lambda}}(\mathcal{T}'))]^2\} = o(1/n). \tag{4}$$

This *loss stability* assumption is rather mild, see Bayle et al. (2020) for an extensive discussion. Further, define the risk $R(g) = \mathbb{E}[\ell(\boldsymbol{Z}, g)]$ and assume that for every $\boldsymbol{\lambda} \in \Lambda$, there is a prediction rule $g_{\boldsymbol{\lambda}}^*$ such that

$$\max_{\boldsymbol{\lambda} \in \Lambda} \mathbb{E}\left[|R(g_{\boldsymbol{\lambda}}(\mathcal{T})) - R(g_{\boldsymbol{\lambda}}^*)|\right] = o(1/\sqrt{n}). \tag{5}$$

This assumption requires $g_{\boldsymbol{\lambda}}(\mathcal{T})$ to converge to some fixed prediction rule sufficiently fast and serves as a reasonable working condition for our purposes. It is satisfied, for example, when $\ell$ is the square loss and $g_{\boldsymbol{\lambda}}$ is an empirical risk minimizer over a hypothesis class $\mathcal{G}_{\boldsymbol{\lambda}}$ with finite VC-dimension. For further examples, see, e.g., Bousquet & Zhivotovskiy (2021), van Erven et al. (2015), and references therein. The assumption could be relaxed, but this would lead to a more complicated limiting distribution but with the same essential interpretation.

**Theorem C.1.** *Under assumptions* (4) *and* (5)*, it holds*

$$\sqrt{n}\left(\widehat{\mu}(\boldsymbol{\lambda}_j) - \mu(\boldsymbol{\lambda}_j)\right)_{j=1}^J \to_d \mathcal{N}(0, \Sigma),$$

*where*

$$\Sigma_{j,j'} = \tau_{i,j,M} \lim_{n \to \infty} \mathsf{Cov}[\bar{\ell}_n(\boldsymbol{Z}, \boldsymbol{\lambda}_j), \bar{\ell}_n(\boldsymbol{Z}, \boldsymbol{\lambda}_{j'})],$$

$$\tau_{j,j',M} = \lim_{n \to \infty} \frac{1}{nM^2\alpha^2} \sum_{i=1}^n \sum_{m=1}^M \sum_{m'=1}^M \Pr(i \in \mathcal{I}_{m,j} \cap \mathcal{I}_{m',j'}).$$

*Proof.* Define

$$\widetilde{\mu}(\boldsymbol{\lambda}_j) = \frac{1}{M} \sum_{m=1}^M \mathbb{E}[L(\mathcal{V}_{m,j}, g_{\boldsymbol{\lambda}_j}(\mathcal{T}_{m,j})) \mid \mathcal{T}_{m,j}].$$

By the triangle inequality (first and second step), Jensen's inequality (third step), and (5) (last step),

$$\begin{aligned}
&\mathbb{E}[|\widetilde{\mu}(\boldsymbol{\lambda}_j) - \mu(\boldsymbol{\lambda}_j)|] \\
&\leq \max_{1 \leq m \leq M} \mathbb{E}\left[\left|\mathbb{E}[L(\mathcal{V}_{m,j}, g_{\boldsymbol{\lambda}_j}(\mathcal{T}_{m,j})) \mid \mathcal{T}_{m,j}] - \mathbb{E}[L(\mathcal{V}_{m,j}, g_{\boldsymbol{\lambda}_j}(\mathcal{T}_{m,j}))]\right|\right] \\
&\leq \max_{1 \leq m \leq M} \mathbb{E}\left[\left|\mathbb{E}[L(\mathcal{V}_{m,j}, g_{\boldsymbol{\lambda}_j}(\mathcal{T}_{m,j})) \mid \mathcal{T}_{m,j}] - \mathbb{E}[L(\mathcal{V}_{m,j}, g_{\boldsymbol{\lambda}_j}^*)]\right|\right] \\
&\quad + \max_{1 \leq m \leq M} \mathbb{E}\left[\left|\mathbb{E}[L(\mathcal{V}_{m,j}, g_{\boldsymbol{\lambda}_j}(\mathcal{T}_{m,j}))] - \mathbb{E}[L(\mathcal{V}_{m,j}, g_{\boldsymbol{\lambda}_j}^*)]\right|\right] \\
&\leq 2 \max_{1 \leq m \leq M} \mathbb{E}\left[\left|\mathbb{E}[L(\mathcal{V}_{m,j}, g_{\boldsymbol{\lambda}_j}(\mathcal{T}_{m,j})) \mid \mathcal{T}_{m,j}] - \mathbb{E}[L(\mathcal{V}_{m,j}, g_{\boldsymbol{\lambda}_j}^*)]\right|\right] \\
&= 2 \max_{1 \leq m \leq M} \mathbb{E}\left[\left|R(g_{\boldsymbol{\lambda}_j}(\mathcal{T}_{m,j})) - R(g_{\boldsymbol{\lambda}_j}^*)\right|\right] \\
&= o(1/\sqrt{n}).
\end{aligned}$$

Next, assumption (4) together with Theorem 2 and Proposition 3 of Bayle et al. (2020) yield

$$\sqrt{n}\left(\widehat{\mu}(\boldsymbol{\lambda}_j) - \widetilde{\mu}(\boldsymbol{\lambda}_j)\right) - \frac{1}{M} \sum_{m=1}^M \frac{1}{\alpha\sqrt{n}} \sum_{i \in \mathcal{I}_{m,j}} \bar{\ell}_n(\boldsymbol{Z}_i, \boldsymbol{\lambda}_j) \to_p 0.$$

Now rewrite

$$\frac{1}{M\alpha\sqrt{n}} \sum_{m=1}^{M} \sum_{i \in \mathcal{I}_{m,j}} \bar{\ell}_n(\mathbf{Z}_i, \boldsymbol{\lambda}_j) = \frac{1}{M\alpha\sqrt{n}} \sum_{i=1}^{n} \underbrace{\sum_{m=1}^{M} \mathbb{1}(i \in \mathcal{I}_{m,j}) \bar{\ell}_n(\mathbf{Z}_i, \boldsymbol{\lambda}_j)}_{:=\xi_{i,n}^{(j)}}.$$

The sequence $(\boldsymbol{\xi}_{i,n})_{i=1}^{n} = (\xi_{i,n}^{(j)}, \ldots, \xi_{i,n}^{(j)})_{i=1}^{n}$ is a triangular array of independent, centered, and bounded random vectors. Because $\mathbb{1}(\mathbf{Z}_i \in \mathcal{V}_{m,j})$ and $\mathbf{Z}_i$ are independent, it holds

$$\mathsf{Cov}(\xi_{i,n}^{(j)}, \xi_{i,n}^{(j')}) = \sum_{m=1}^{M} \sum_{m'=1}^{M} \mathbb{E}[\mathbb{1}(i \in \mathcal{I}_{m,j} \cap \mathcal{I}_{m',j'})]\mathbb{E}[\bar{\ell}_n(\mathbf{Z}_i, \boldsymbol{\lambda}_j)\bar{\ell}_n(\mathbf{Z}_i, \boldsymbol{\lambda}_{j'})],$$

so

$$\lim_{n\to\infty} \mathsf{Cov}\left[\frac{1}{M\alpha\sqrt{n}} \sum_{i=1}^{n} \xi_{i,n}^{(j)}, \frac{1}{M\alpha\sqrt{n}} \sum_{i=1}^{n} \xi_{i,n}^{(j')}\right] = \lim_{n\to\infty} \frac{1}{nM^2\alpha^2} \sum_{i=1}^{n} \mathsf{Cov}\left[\xi_{i,n}^{(j)}, \xi_{i,n}^{(j')}\right] = \Sigma_{j,j'}.$$

Now the result follows from Lindeberg's central limit theorem for triangular arrays (e.g., van der Vaart, 2000, Proposition 2.27). $\qquad\square$

## C.2 Proof of Theorem 2.2

We want to bound the probability that $\mu(\hat{\boldsymbol{\lambda}}) - \mu(\boldsymbol{\lambda}^*)$ is large. For some $\delta > 0$, define the set of 'good' hyperparameters

$$\Lambda_\delta = \{\boldsymbol{\lambda}_j : \mu(\boldsymbol{\lambda}_j) - \mu(\boldsymbol{\lambda}^*) \le \delta\}.$$

Now

$$\begin{aligned}
\Pr\left(\mu(\widehat{\boldsymbol{\lambda}}) - \mu(\boldsymbol{\lambda}^*) > \delta\right) &= \Pr\left(\widehat{\boldsymbol{\lambda}} \notin \Lambda_\delta\right) \\
&= \Pr\left(\min_{\boldsymbol{\lambda}\notin\Lambda_\delta} \widehat{\mu}(\boldsymbol{\lambda}) < \min_{\boldsymbol{\lambda}\in\Lambda_\delta} \widehat{\mu}(\boldsymbol{\lambda})\right) \\
&\le \Pr\left(\min_{\boldsymbol{\lambda}\notin\Lambda_\delta} \widehat{\mu}(\boldsymbol{\lambda}) < \min_{\boldsymbol{\lambda}\in\Lambda_{\delta/2}} \widehat{\mu}(\boldsymbol{\lambda})\right) \\
&= \Pr\left(\min_{\boldsymbol{\lambda}\notin\Lambda_\delta} \mu(\boldsymbol{\lambda}) + \epsilon(\boldsymbol{\lambda}) < \min_{\boldsymbol{\lambda}\in\Lambda_{\delta/2}} \mu(\boldsymbol{\lambda}) + \epsilon(\boldsymbol{\lambda})\right) \\
&\le \Pr\left(\delta + \min_{\boldsymbol{\lambda}\notin\Lambda_\delta} \epsilon(\boldsymbol{\lambda}) < \delta/2 + \min_{\boldsymbol{\lambda}\in\Lambda_{\delta/2}} \epsilon(\boldsymbol{\lambda})\right) \\
&= \Pr\left(\min_{\boldsymbol{\lambda}\notin\Lambda_\delta} \epsilon(\boldsymbol{\lambda}) - \min_{\boldsymbol{\lambda}\in\Lambda_{\delta/2}} \epsilon(\boldsymbol{\lambda}) < -\delta/2\right) \\
&= \Pr\left(\max_{\boldsymbol{\lambda}\notin\Lambda_\delta} \epsilon(\boldsymbol{\lambda}) - \max_{\boldsymbol{\lambda}\in\Lambda_{\delta/2}} \epsilon(\boldsymbol{\lambda}) > \delta/2\right). \qquad (\epsilon \overset{d}{=} -\epsilon)
\end{aligned}$$

There is a tension between the two maxima. The more $\boldsymbol{\lambda}$'s there are in $\Lambda_{\delta/2}$ and the less they are correlated, the more likely it is to find one $\epsilon(\boldsymbol{\lambda})$ that is large. This makes the probability small. However, the less $\epsilon$ is correlated, the larger is $\max_{\boldsymbol{\lambda}\notin\Lambda_\delta} \epsilon(\boldsymbol{\lambda})$, making the probability large. To formalize this, use the Gaussian concentration inequality (Talagrand, 2005, Lemma 2.1.3):

$$\begin{aligned}
&\Pr\left(\max_{\boldsymbol{\lambda}\notin\Lambda_\delta} \epsilon(\boldsymbol{\lambda}) - \max_{\boldsymbol{\lambda}\in\Lambda_{\delta/2}} \epsilon(\boldsymbol{\lambda}) > \delta/2\right) \\
&\le \Pr\left(2\left|\max_{\boldsymbol{\lambda}\in\Lambda} \epsilon(\boldsymbol{\lambda}) - \mathbb{E}\left[\max_{\boldsymbol{\lambda}\in\Lambda} \epsilon(\boldsymbol{\lambda})\right]\right| > \delta/2 - \mathbb{E}\left[\max_{\boldsymbol{\lambda}\in\Lambda_{\delta/2}} \epsilon(\boldsymbol{\lambda})\right] + \mathbb{E}\left[\max_{\boldsymbol{\lambda}\notin\Lambda_\delta} \epsilon(\boldsymbol{\lambda})\right]\right) \\
&\le 2\exp\left\{-\frac{\left(\delta/2 - \mathbb{E}\left[\max_{\boldsymbol{\lambda}\in\Lambda_{\delta/2}} \epsilon(\boldsymbol{\lambda})\right] + \mathbb{E}\left[\max_{\boldsymbol{\lambda}\notin\Lambda_\delta} \epsilon(\boldsymbol{\lambda})\right]\right)^2}{8\sigma^2}\right\},
\end{aligned}$$

provided $\delta/2 - \mathbb{E}\left[\max_{\boldsymbol{\lambda}\in\Lambda_{\delta/2}} \epsilon(\boldsymbol{\lambda})\right] + \mathbb{E}\left[\max_{\boldsymbol{\lambda}\notin\Lambda_\delta} \epsilon(\boldsymbol{\lambda})\right] \ge 0$. We bound the two maxima separately.

**Lower Bound for Maximum over the Good Set**

Recall the definition of $m$ right before Theorem 2.2 and observe

$$\Lambda_{\delta/2} = \{\boldsymbol{\lambda}\colon \mu(\boldsymbol{\lambda}) - \mu(\boldsymbol{\lambda}^*) \leq \delta/2\} \supset \{\boldsymbol{\lambda}\colon m\|\boldsymbol{\lambda} - \boldsymbol{\lambda}^*\|^2 \leq \delta/2\} = \{\boldsymbol{\lambda}\colon \|\boldsymbol{\lambda} - \boldsymbol{\lambda}^*\| \leq (\delta/2m)^{1/2}\}$$
$$= B(\boldsymbol{\lambda}^*, (\delta/2m)^{1/2}).$$

Pack the ball $B(\boldsymbol{\lambda}^*, (\delta/2m)^{1/2})$ with smaller balls with radius $\eta$. We can always construct such a packing with at least $(\delta/2m\eta^2)^{d/2}$ elements. By assumption, each small ball contains at least one element of $\Lambda$. Pick one element from each small ball and collect them into the set $\Lambda'_{\delta/2}$. By construction, $|\Lambda'_{\delta/2}| \geq (\delta/2m\eta^2)^{d/2}$ and

$$\min_{\boldsymbol{\lambda}\neq\boldsymbol{\lambda}'\in\Lambda'_{\delta/2}|} \|\boldsymbol{\lambda} - \boldsymbol{\lambda}'\| \geq \eta.$$

Sudakov's minoration principle (e.g., Wainwright, 2019, Theorem 5.30) gives

$$\mathbb{E}\left[\max_{\boldsymbol{\lambda}\in\Lambda_{\delta/2}} \epsilon(\boldsymbol{\lambda})\right] \geq \frac{1}{2}\sqrt{\log|\Lambda'_{\delta/2}|} \min_{\{\boldsymbol{\lambda}\neq\boldsymbol{\lambda}'\}\cap\Lambda'_{\delta/2}} \sqrt{\mathsf{Var}\left[\epsilon(\boldsymbol{\lambda}) - \epsilon(\boldsymbol{\lambda}')\right]}$$
$$\geq \frac{1}{2}\sqrt{\log|\Lambda'_{\delta/2}|} \min_{\|\boldsymbol{\lambda}-\boldsymbol{\lambda}'\|\geq\eta} \sqrt{\mathsf{Var}\left[\epsilon(\boldsymbol{\lambda}) - \epsilon(\boldsymbol{\lambda}')\right]}.$$

In general,

$$\mathsf{Var}\left[\epsilon(\boldsymbol{\lambda}) - \epsilon(\boldsymbol{\lambda}')\right]$$
$$= K(\boldsymbol{\lambda}, \boldsymbol{\lambda}) + K(\boldsymbol{\lambda}', \boldsymbol{\lambda}') - 2\tau^2 K(\boldsymbol{\lambda}, \boldsymbol{\lambda}')$$
$$= (1 - \tau^2)[K(\boldsymbol{\lambda}, \boldsymbol{\lambda}) + K(\boldsymbol{\lambda}', \boldsymbol{\lambda}')] + \tau^2[K(\boldsymbol{\lambda}, \boldsymbol{\lambda}) - K(\boldsymbol{\lambda}, \boldsymbol{\lambda}')] + \tau^2[K(\boldsymbol{\lambda}', \boldsymbol{\lambda}') - K(\boldsymbol{\lambda}, \boldsymbol{\lambda}')]$$
$$\geq 2\underline{\sigma}^2(1 - \tau^2).$$

Hence, we have

$$\min_{\|\boldsymbol{\lambda}-\boldsymbol{\lambda}'\|\geq\eta} \mathsf{Var}\left[\epsilon(\boldsymbol{\lambda}) - \epsilon(\boldsymbol{\lambda}')\right] \geq 2\underline{\sigma}^2(1 - \tau^2),$$

which implies

$$\mathbb{E}\left[\max_{\boldsymbol{\lambda}\in\Lambda_{\delta/2}} \epsilon(\boldsymbol{\lambda})\right] \geq \frac{1}{2}\underline{\sigma}\sqrt{d}\sqrt{1 - \tau^2}\sqrt{\log(\delta/2m\eta^2)} =: \underline{\sigma}\sqrt{d}A(\tau, \delta)/2.$$

**Upper Bound for Maximum over the Bad Set**

Dudley's entropy bound (e.g., Giné & Nickl, 2016, Theorem 2.3.6) gives

$$\mathbb{E}\left[\max_{\boldsymbol{\lambda}\notin\Lambda_\delta} \epsilon(\boldsymbol{\lambda})\right] \leq 12 \int_0^\infty \sqrt{\log N(s)}ds,$$

where $N(s)$ is the minimum number of points $\boldsymbol{\lambda}_1, \ldots, \boldsymbol{\lambda}_{N(s)}$ such that

$$\sup_{\boldsymbol{\lambda}\in\Lambda} \min_{1\leq k\leq N(s)} \sqrt{\mathsf{Var}\left[\epsilon(\boldsymbol{\lambda}) - \epsilon(\boldsymbol{\lambda}_k)\right]} \leq s.$$

Note that

$$\sup_{\boldsymbol{\lambda},\boldsymbol{\lambda}'\in\Lambda} \sqrt{\mathsf{Var}\left[\epsilon(\boldsymbol{\lambda}) - \epsilon(\boldsymbol{\lambda}')\right]} \leq 2\sigma,$$

so $N(s) = 1$ for all $s \geq 2\sigma$. For $s^2 \leq 4\sigma^2(1 - \tau^2)$, we can use the trivial bound $N(s) \leq J$. For $s^2 > 4\sigma^2(1 - \tau^2)$, cover $\Lambda$ with $\ell_2$-balls of size $(s/2\sigma\tau\kappa)$. We can do this with less than $N(s) \leq (6\sigma\kappa/s)^d \vee 1$ such balls. Let $\boldsymbol{\lambda}_1, \ldots, \boldsymbol{\lambda}_N$ be the centers of these balls. In general, it holds

$$\mathsf{Var}\left[\epsilon(\boldsymbol{\lambda}) - \epsilon(\boldsymbol{\lambda}')\right]$$
$$= K(\boldsymbol{\lambda}, \boldsymbol{\lambda}) + K(\boldsymbol{\lambda}', \boldsymbol{\lambda}') - 2\tau^2 K(\boldsymbol{\lambda}, \boldsymbol{\lambda}')$$
$$= (1 - \tau^2)[K(\boldsymbol{\lambda}, \boldsymbol{\lambda}) + K(\boldsymbol{\lambda}', \boldsymbol{\lambda}')] + \tau^2[K(\boldsymbol{\lambda}, \boldsymbol{\lambda}) - K(\boldsymbol{\lambda}, \boldsymbol{\lambda}')] + \tau^2[K(\boldsymbol{\lambda}', \boldsymbol{\lambda}') - K(\boldsymbol{\lambda}, \boldsymbol{\lambda}')]$$
$$\leq 2(1 - \tau^2)\sigma^2 + 2\tau^2\sigma^2\kappa^2\|\boldsymbol{\lambda} - \boldsymbol{\lambda}'\|^2.$$

For $s^2 > 4\sigma^2(1-\tau^2)$, we thus have

$$\sup_{\boldsymbol{\lambda}\in\Lambda} \min_{1\leq k\leq N(s)} \mathsf{Var}\left[\epsilon(\boldsymbol{\lambda}) - \epsilon(\boldsymbol{\lambda}_k)\right] \leq \sup_{\|\boldsymbol{\lambda}-\boldsymbol{\lambda}'\|_2\leq(s/2\tau\sigma\kappa)^2} \mathsf{Var}\left[\epsilon(\boldsymbol{\lambda}) - \epsilon(\boldsymbol{\lambda}')\right]$$
$$\leq 2(1-\tau^2)\sigma^2 + 2\tau^2\sigma^2\kappa^2(s/2\tau\sigma\kappa)^2$$
$$\leq s^2,$$

as desired. Now decompose the integral

$$\int_0^\infty \sqrt{\log N(s)}ds = \int_0^{2\sigma\sqrt{1-\tau^2}} \sqrt{\log N(s)}ds + \int_{2\sigma\sqrt{1-\tau^2}}^{2\sigma} \sqrt{\log N(s)}ds$$
$$\leq 2\sigma\sqrt{d}\sqrt{1-\tau^2}\sqrt{\log J} + \int_{2\sigma\sqrt{1-\tau^2}}^{2\sigma} \sqrt{\log N(s)}ds.$$

For the second term, compute

$$\int_{\sigma\sqrt{1-\tau^2}}^{2\sigma} \sqrt{\log N(s)}ds \leq \sqrt{d}\int_{2\sigma\sqrt{1-\tau^2}}^{2\sigma} \sqrt{\log(6\sigma\kappa/s)_+}\,ds$$
$$= \sigma\sqrt{d}\int_{2\sqrt{1-\tau^2}}^{2} \sqrt{\log(6\kappa/s)_+}\,ds$$
$$\leq \sigma\sqrt{d}\left(\int_0^2 \log(6\kappa/s)_+\,ds\right)^{1/2}\left(2(1-\sqrt{1-\tau^2})\right)^{1/2}$$
$$= \sigma\sqrt{d}\sqrt{2 + 2\log(3\kappa)_+}\left(2(1-\sqrt{1-\tau^2})\right)^{1/2}$$
$$= 2\sigma\sqrt{d}\sqrt{1 + \log(3\kappa)_+}\frac{\tau}{(1+\sqrt{1-\tau^2})^{1/2}}$$
$$\leq 2\sigma\sqrt{d}\tau\sqrt{1 + \log(3\kappa)_+}.$$

We have shown that

$$\mathbb{E}\left[\max_{\boldsymbol{\lambda}\notin\Lambda_\delta} \epsilon(\boldsymbol{\lambda})\right] \leq 24\sigma\sqrt{d}\left[\sqrt{1-\tau^2}\sqrt{\log J} + \tau\sqrt{1 + \log(3\kappa)_+}\right] =: \sigma\sqrt{d}B(\tau)/4.$$

**Integrating Probabilities**

Summarizing the two previous steps, we have

$$\Pr\left(\mu(\widehat{\boldsymbol{\lambda}}) - \mu(\boldsymbol{\lambda}^*) > \delta\right) \leq 2\exp\left\{-\frac{\left(\delta - \sigma\sqrt{d}[B(\tau) - A(\tau,\delta)]\right)^2}{36\sigma^2}\right\},$$

provided $t \geq \sigma\sqrt{d}[B(\tau) - A(\tau,\delta)]$. Now for any $s \geq 0$ and $t \geq 2e^{s^2}m\eta^2$, it holds

$$A(\tau, s) \geq (\underline{\sigma}/\sigma)\sqrt{1-\tau^2}s =: A(\tau)s.$$

In particular, if

$$t \geq 2e^{s^2}m\eta^2 + \sigma\sqrt{d}[B(\tau) - A(\tau)s] =: C,$$

we have

$$\Pr\left(\mu(\widehat{\boldsymbol{\lambda}}) - \mu(\boldsymbol{\lambda}^*) > \delta\right) \leq 4\exp\left\{-\frac{\left(\delta - \sigma\sqrt{d}[B(\tau) - A(\tau)s]\right)^2}{36\sigma^2}\right\}.$$

Integrating the probability gives

$$
\begin{aligned}
\mathbb{E}[\mu(\widehat{\boldsymbol{\lambda}}) - \mu(\boldsymbol{\lambda}^*)] &= \int_0^\infty \Pr\left(\mu(\widehat{\boldsymbol{\lambda}}) - \mu(\boldsymbol{\lambda}^*) > \delta\right) d\delta \\
&= \int_0^C \Pr\left(\mu(\widehat{\boldsymbol{\lambda}}) - \mu(\boldsymbol{\lambda}^*) > \delta\right) d\delta + \int_C^\infty \Pr\left(\mu(\widehat{\boldsymbol{\lambda}}) - \mu(\boldsymbol{\lambda}^*) > \delta\right) d\delta \\
&\leq C + \int_C^\infty \exp\left\{-\frac{\left(\delta - \sigma\sqrt{d}[B(\tau) - A(\tau)s]\right)^2}{36\sigma^2}\right\} d\delta \\
&\leq C + \sqrt{36}\sigma \\
&= 2e^{s^2}m\eta^2 + \sigma\sqrt{d}[B(\tau) - A(\tau)s] + 6\sigma.
\end{aligned}
$$

**Simplifying**

The bound can be optimized with respect to $s$, but the solution involves the Lambert $W$-function, which has no analytical expression. Instead choose $s$ for simplicity as

$$
s = \sqrt{\log\left(\frac{\sigma}{2m\eta^2}\right)_+}.
$$

which gives

$$
\mathbb{E}[\mu(\widehat{\boldsymbol{\lambda}}) - \mu(\boldsymbol{\lambda}^*)] \leq \sigma\sqrt{d}\left[8 + B(\tau) - A(\tau)\sqrt{\log\left(\frac{\sigma}{2m\eta^2}\right)}\right]. \qquad \square
$$

## D  Additional Results on the Density of Random HPC Grids

**Lemma D.1.** *Suppose that the $J$ elements in $\Lambda$ are drawn independently from a continuous density $p$ with $c := \min_{\|\boldsymbol{\lambda}\|\leq 1} p(\boldsymbol{\lambda}) > 0$. Then with probability at least $1 - \delta$,*

$$
\eta \lesssim \left(\sqrt{\log(1/\delta)/J}\right)^{1/d},
$$

*and with probability 1,*

$$
\eta \lesssim \left(\sqrt{\log(J)/J}\right)^{1/d},
$$

*for all $J$ sufficiently large.*

*Proof.* We want to bound the probability that there is a $\boldsymbol{\lambda}$ such that $|B(\boldsymbol{\lambda}, \eta) \cap \Lambda| = 0$. In what follows $\boldsymbol{\lambda}$ is silently understood to have norm bounded by 1. Let $\widetilde{\boldsymbol{\lambda}}_1, \ldots, \widetilde{\boldsymbol{\lambda}}_N$ the centers of $\eta/2$-balls covering $\{\|\boldsymbol{\lambda}\| \leq 1\}$, for which we may assume $N \leq (6/\eta)^d$. For $\widetilde{\boldsymbol{\lambda}}_k$ the closest center to $\boldsymbol{\lambda}$, it holds

$$
\|\boldsymbol{\lambda}' - \boldsymbol{\lambda}\| \leq \|\boldsymbol{\lambda}' - \widetilde{\boldsymbol{\lambda}}_k\| + \|\widetilde{\boldsymbol{\lambda}}_k - \boldsymbol{\lambda}\| \leq \|\boldsymbol{\lambda}' - \widetilde{\boldsymbol{\lambda}}_k\| + \eta/2,
$$

so $\|\boldsymbol{\lambda}' - \widetilde{\boldsymbol{\lambda}}_k\| \leq \eta/2$ implies $\|\boldsymbol{\lambda}' - \boldsymbol{\lambda}\| \leq \eta$. We thus have

$$
\begin{aligned}
\Pr(\exists \boldsymbol{\lambda} \colon |B(\boldsymbol{\lambda}, \eta) \cap \Lambda| = 0) &= \Pr\left(\inf_{\boldsymbol{\lambda}} \sum_{i=1}^J \mathbb{1}\{\|\boldsymbol{\lambda}_i - \boldsymbol{\lambda}\| \leq \eta\} \leq 0\right) \\
&\leq \Pr\left(\min_{1\leq k\leq N} \sum_{i=1}^J \mathbb{1}\{\|\boldsymbol{\lambda}_i - \widetilde{\boldsymbol{\lambda}}_k\| \leq \eta/2\} \leq 0\right).
\end{aligned}
$$

Further

$$\Pr\left(\min_{1\le k\le N}\sum_{i=1}^{J}\mathbb{1}\{\|\boldsymbol{\lambda}_i - \widetilde{\boldsymbol{\lambda}}_k\| \le \eta/2\} \le 0\right)$$

$$= \Pr\left(\max_{1\le k\le N}\sum_{i=1}^{J}-\mathbb{1}\{\|\boldsymbol{\lambda}_i - \widetilde{\boldsymbol{\lambda}}_k\| \le \eta/2\} \ge 0\right)$$

$$\le \Pr\left(\max_{1\le k\le N}\sum_{i=1}^{J}\mathbb{E}\left[\mathbb{1}\{\|\boldsymbol{\lambda}_i - \widetilde{\boldsymbol{\lambda}}_k\| \le \eta/2\}\right] - \mathbb{1}\{\|\boldsymbol{\lambda}_i - \widetilde{\boldsymbol{\lambda}}_k\| \le \eta/2\} \ge J\inf_{\boldsymbol{\lambda}}\mathbb{E}\left[\mathbb{1}\{\|\boldsymbol{\lambda}_i - \boldsymbol{\lambda}\| \le \eta/2\}\right]\right).$$

It holds

$$\mathbb{E}\left[\mathbb{1}\{\|\boldsymbol{\lambda}_i - \boldsymbol{\lambda}\| \le \eta/2\}\right] = \Pr\left(\|\boldsymbol{\lambda}_i - \boldsymbol{\lambda}\| \le \eta/2\right) = \int_{\|\boldsymbol{\lambda}' - \boldsymbol{\lambda}\| \le \eta/2} p(\boldsymbol{\lambda}')d\boldsymbol{\lambda}' \ge c\,\mathrm{vol}(B(0,\eta/2))$$

$$= cv_d(\eta/2)^d,$$

where $v_d = \mathrm{vol}(B(0,1))$. Now the union bound and Hoeffding's inequality give

$$\Pr\left(\min_{1\le k\le N}\sum_{i=1}^{J}\mathbb{1}\{\|\boldsymbol{\lambda}_i - \widetilde{\boldsymbol{\lambda}}_k\| \le \eta/2\} \le 0\right) \le N\exp\left(-\frac{Jc^2 v_d^2(\eta/2)^{2d}}{2}\right)$$

$$\le (6/\eta)^d\exp\left(-\frac{Jc^2 v_d^2(\eta/2)^{2d}}{2}\right).$$

Choosing

$$\eta = 2\left(\sqrt{2\log(3^d\sqrt{J}cv_d/\delta)}/\sqrt{J}cv_d\right)^{1/d}$$

gives

$$\Pr(\exists\boldsymbol{\lambda}\colon |B(\boldsymbol{\lambda},\eta)\cap\Lambda| = 0) \le \delta/\sqrt{2\log(3^d\sqrt{J}cv_d)},$$

which is bounded by $\delta$ when $\sqrt{J} \ge e^{1/2}/3^d cv_d$. Further, setting $\eta = 2(\sqrt{6\log(J)}/\sqrt{J}cv_d)^{1/d}$ gives

$$\Pr\left(\min_{1\le k\le N}\sum_{i=1}^{J}\mathbb{1}\{\|\boldsymbol{\lambda}_i - \widetilde{\boldsymbol{\lambda}}_k\| \le \eta/2\} \le 0\right) \lesssim J^{-5/2},$$

so that

$$\sum_{J=1}^{\infty}\Pr\left(\min_{1\le j\le J}\min_{1\le k\le N}\sum_{i=1}^{j}\mathbb{1}\{\|\boldsymbol{\lambda}_i - \widetilde{\boldsymbol{\lambda}}_k\| \le \eta/2\} \le 0\right)$$

$$\le \sum_{J=1}^{\infty}J\Pr\left(\min_{1\le k\le N}\sum_{i=1}^{J}\mathbb{1}\{\|\boldsymbol{\lambda}_i - \widetilde{\boldsymbol{\lambda}}_k\| \le \eta/2\} \le 0\right)$$

$$\lesssim \sum_{J=1}^{\infty}\frac{1}{J^{3/2}} < \infty.$$

Now the Borel-Cantelli lemma (e.g., Kallenberg, 1997, Theorem 4.18) implies that, with probability 1,

$$|B(\boldsymbol{\lambda},\eta)\cap\Lambda| \ge 1,$$

for all $J$ sufficiently large. $\qquad\square$

# E   Selected Validation Schemes

## E.1   Definition of Index Sets

Recall:

(i) (holdout) Let $M = 1$ and $\mathcal{I}_{1,j} = \mathcal{I}_1$ for all $j = 1, \ldots, J$, and some size-$\lceil \alpha n \rceil$ index set $\mathcal{I}_1$.

(ii) (reshuffled holdout) Let $M = 1$ and $\mathcal{I}_{1,1}, \ldots, \mathcal{I}_{1,J}$ be independently drawn from the uniform distribution over all size-$\lceil \alpha n \rceil$ subsets from $\{1, \ldots, n\}$.

(iii) ($M$-fold CV) Let $\alpha = 1/M$ and $\mathcal{I}_1, \ldots, \mathcal{I}_M$ be a disjoint partition of $\{1, \ldots, n\}$, and $\mathcal{I}_{m,j} = \mathcal{I}_m$ for all $j = 1, \ldots, J$.

(iv) (reshuffled $M$-fold CV) Let $\alpha = 1/M$ and $(\mathcal{I}_{1,j}, \ldots, \mathcal{I}_{M,j}), j = 1, \ldots, J$, be independently drawn from the uniform distribution over disjoint partitions of $\{1, \ldots, n\}$.

(v) ($M$-fold holdout) Let $\mathcal{I}_m, m = 1, \ldots, M$, be independently drawn from the uniform distribution over size-$\lceil \alpha n \rceil$ subsets of $\{1, \ldots, n\}$ and set $\mathcal{I}_{m,j} = \mathcal{I}_m$ for all $m = 1, \ldots, M, j = 1, \ldots, J$.

(vi) (reshuffled $M$-fold holdout) Let $\mathcal{I}_{m,j}, m = 1, \ldots, M, j = 1, \ldots, J$, be independently drawn from the uniform distribution over size-$\lceil \alpha n \rceil$ subsets of $\{1, \ldots, n\}$.

## E.2   Derivation of Reshuffling Parameters in Limiting Distribution

Recall

$$\tau_{i,j,M} = \frac{1}{nM^2\alpha^2} \sum_{s=1}^{n} \sum_{m=1}^{M} \sum_{m'=1}^{M} \Pr(s \in \mathcal{I}_{m,i} \cap \mathcal{I}_{m',j}).$$

For all schemes in the proposition, the probabilities are independent of the index $s$, so the average over $s = 1, \ldots, n$ can be omitted. We now verify the constants $\sigma, \tau$ from Table 1.

(i) It holds

$$\Pr(s \in \mathcal{I}_{1,i} \cap \mathcal{I}_{1,j}) = \Pr(s \in \mathcal{I}_1) = \alpha.$$

Hence,

$$\tau_{i,j,1} = 1/\alpha = 1/\alpha \times 1 = \sigma^2 \times \tau^2.$$

(ii) (reshuffled holdout) This is a special case of part (vi) with $M = 1$.

(iii) ($M$-fold CV) It holds

$$\Pr(s \in \mathcal{I}_{m,i} \cap \mathcal{I}_{m',j}) = \Pr(s \in \mathcal{I}_m \cap \mathcal{I}_{m'}) = \begin{cases} 1/M, & m = m', \\ 0, & m \neq m'. \end{cases}$$

Only $M$ probabilities in the double sum are non-zero, whence

$$\tau_{i,j,M} = \frac{1}{M^2\alpha^2} \times M/M = 1/\alpha^2 M^2 = 1 \times 1 = \sigma^2 \times \tau^2,$$

where we used $\alpha = 1/M$.

(iv) (reshuffled $M$-fold CV) It holds

$$\Pr(s \in \mathcal{I}_{m,i} \cap \mathcal{I}_{m',j}) = \begin{cases} 1/M, & m = m', i = j \\ 0, & m \neq m', i = j \\ 1/M^2, & m = m', i \neq j \\ 1/M^2, & m \neq m', i \neq j. \end{cases}$$

For $i = j$, only $M$ probabilities in the double sum are non-zero. Also using $\alpha = 1/M$, we get

$$\tau_{i,j,M} = \frac{1}{M^2\alpha^2} \times M \times 1/M = 1 = \sigma^2.$$

For $i \neq j$,

$$\tau_{i,j,M} = \frac{1}{M^2\alpha^2} \times M^2 \times 1/M^2 = 1 \times 1 = \sigma^2 \times \tau^2.$$

(v) ($M$-fold holdout) It holds

$$\Pr(s \in \mathcal{I}_{m,i} \cap \mathcal{I}_{m',j}) = \Pr(s \in \mathcal{I}_m \cap \mathcal{I}_{m'}) = \begin{cases} \alpha, & m = m', \\ \alpha^2, & \text{else.} \end{cases}$$

This gives

$$\tau_{i,j,M} = \frac{1}{M^2\alpha^2} \times [M \times \alpha + (M-1)M \times \alpha^2] = [1/\alpha M + (M-1)/M] \times 1 = \sigma^2 \times \tau^2.$$
for all $i, j$.

(vi) (reshuffled $M$-fold holdout) It holds

$$\Pr(s \in \mathcal{I}_{m,i} \cap \mathcal{I}_{m',j}) = \begin{cases} \alpha, & m = m', i = j \\ \alpha^2, & \text{else.} \end{cases}$$

For $i = j$, this gives

$$\tau_{i,j,M} = \frac{1}{M^2\alpha^2} \times [M \times \alpha + (M-1)M \times \alpha^2] = 1/\alpha M + (M-1)/M.$$

For $i \neq j$,

$$\tau_{i,j,M} = \frac{1}{M^2\alpha^2} \times (M^2 \times \alpha^2) = 1.$$

This implies that (1) holds with $\sigma^2 = 1/M\alpha + (M-1)/M, \tau^2 = 1/(1/M\alpha + (M-1)/M)$.

**Remark E.1.** *Although not technically covered by Theorem 2.1, performing independent bootstraps for each $\boldsymbol{\lambda}_j$ correspond to reshuffled $n$-fold holdout with $\alpha = 1/n$. Accordingly, $\sigma \approx \sqrt{2}$ and $\tau \approx \sqrt{1/2}$.*

## F  Details Regarding Benchmark Experiments

### F.1  Datasets

We list all datasets used in the benchmark experiments in Table 4.

Table 4: List of datasets used in benchmark experiments. All information can be found on OpenML (Vanschoren et al., 2014).

| OpenML Dataset ID | Dataset Name | Size ($n \times p$) |
|---|---|---|
| 23517 | numerai28.6 | $96320 \times 21$ |
| 1169 | airlines | $539383 \times 7$ |
| 41147 | albert | $425240 \times 78$ |
| 4135 | Amazon_employee_access | $32769 \times 9$ |
| 1461 | bank-marketing | $45211 \times 16$ |
| 1590 | adult | $48842 \times 14$ |
| 41150 | MiniBooNE | $130064 \times 50$ |
| 41162 | kick | $72983 \times 32$ |
| 42733 | Click_prediction_small | $39948 \times 11$ |
| 42742 | porto-seguro | $595212 \times 57$ |

Note that datasets serve as data generating processes (DGPs; Hothorn et al., 2005). As we are mostly concerned with the actual generalization performance of the final best HPC found during HPO based on validation performance we rely on a comparably large held out test set that is not used during HPO. We therefore use 5000 data points sampled from a DGP as an outer test set. To further be able to measure the generalization performance robustly for varying data sizes available during HPO, we construct concrete tasks based on the DGPs by sampling subsets of (train_valid; $n$) size 500, 1000 and 5000 from the DGPs. This results in 30 tasks in total (10 DGPS $\times$ 3 train_valid sizes). For more details and the concrete implementation of this procedure, see Appendix F.3. We also collected another 5000 data points as an external validation set, but did not use it. Therefore, we had to tighten the restriction to 10000 data points mentioned in the main paper to 15000 data points as the lower bound on data points. To allow for stronger variation over different replications, we decided to use 20000 as the final lower bound.

## F.2 Learning Algorithms

Here we briefly present training pipeline details and search spaces of the learning algorithms used in our benchmark experiments.

The funnel-shaped MLP is based on sklearn's MLP Classifier and is constructed in the following way: The hidden layer size for each layer is determined by `num_layers` and `max_units`. We start with `max_units` and half the number of units for every subsequent layer to create a funnel. `max_batch_size` is the largest power of 2 that is smaller than the number of training samples available. We use ReLU as activation function and train the network optimizing logloss as a loss function via SGD using a constant learning rate and Nesterov momentum for 100 epochs. Table 5 lists the search space (inspired from Zimmer et al. (2021)) used during HPO.

The Elastic Net is based on sklearn's Logistic Regression Classifier. We train it for a maximum of 1000 iterations using the "saga" solver. Table 6 lists the search space used during HPO.

The XGBoost and CatBoost search spaces are listed in Table 7 and Table 8, both inspired from their search spaces used in McElfresh et al. (2023).

For both the Elastic Net and Funnel MLP, missing values are imputed in the preprocessing pipeline (mean imputation for numerical features and adding a new level for categorical features). Categorical features are target encoded in a cross-validated manner using a 5-fold CV. Features are then scaled to zero mean and unit variance via a standard scaler. For XGBoost, we impute missing values for categorical features (adding a new level) and target encode them in a cross-validated manner using a 5-fold CV. For CatBoost, no preprocessing is performed.

XGBoost and CatBoost models are trained for 2000 iterations and stop early if the validation loss (using the default internal loss function used during training, i.e., logloss) does not improve over a horizon of 20 iterations. For retraining the best configuration on the whole train and validation data, the number of boosting iterations is set to the number of iterations used to find the best validation performance prior to the stopping mechanism taking action.[7]

## F.3 Exact Implementation

In the following, we outline the exact implementation of performing one HPO run for a given learning algorithm on a concrete task (dataset $\times$ `train_valid` size) and a given resampling. We release all code to replicate benchmark results and reproduce our analyses via https://github.com/slds-lmu/paper_2024_reshuffling. For a given replication (in total 10):

1. We sample (without replacement) `train_valid` size (500, 1000 or 5000 points) and `test` size (always 5000) points from the DGP (i.e. a concrete dataset in Table 4). These are shared for every learning algorithm (i.e. all learning algorithms are evaluated on the same data).

2. A given HPC is evaluated in the following way:

   - The resampling operates on the train validation[8] set of size `train_valid`.
   - The learning algorithm is configured by the HPC.
   - The learning algorithm is trained on training splits and evaluated on validation splits according to the resampling strategy. In case reshuffling is turned on, the training and validation splits are recreated for every HPO. We compute the Accuracy, ROC AUC and logloss when using a random search and compute ROC AUC when using HEBO or SMAC3 and average performance over all folds for resamplings involving multiple folds.
   - For each HPC we then always re-train the model on all `train_valid` data being available and evaluate the model on the held-out `test` set to compute an outer estimate of generalization performance for each HPC (regardless of whether it is the incumbent for a given iteration or not).

---

[7]For CV and repeated holdout we take the average number of boosting iterations over the models trained on the different folds.

[8]With train validation we refer to all data being available during HPO which is then further split by a resampling into train and validation sets.

Table 5: Search Space for Funnel-Shaped MLP Classifier.

| Parameter | Type | Range | Log |
|-----------|------|-------|-----|
| num_layers | Int. | 1 to 5 | No |
| max_units | Int. | 64, 128, 256, 512 | No |
| learning_rate | Num. | $1 \times 10^{-4}$ to $1 \times 10^{-1}$ | Yes |
| batch_size | Int. | 16, 32, ..., max_batch_size | No |
| momentum | Num. | 0.1 to 0.99 | No |
| alpha | Num. | $1 \times 10^{-6}$ to $1 \times 10^{-1}$ | Yes |

Table 6: Search Space for Elastic Net Classifier.

| Parameter | Type | Range | Log |
|-----------|------|-------|-----|
| C | Num. | $1 \times 10^{-6}$ to $1 \times 10^{4}$ | Yes |
| l1_ratio | Num. | 0.0 to 1.0 | No |

Table 7: Search Space for XGBoost Classifier.

| Parameter | Type | Range | Log |
|-----------|------|-------|-----|
| max_depth | Int. | 2 to 12 | Yes |
| alpha | Num. | $1 \times 10^{-8}$ to 1.0 | Yes |
| lambda | Num. | $1 \times 10^{-8}$ to 1.0 | Yes |
| eta | Num. | 0.01 to 0.3 | Yes |

Table 8: Search Space for CatBoost Classifier.

| Parameter | Type | Range | Log |
|-----------|------|-------|-----|
| learning_rate | Num. | 0.01 to 0.3 | Yes |
| depth | Int. | 2 to 12 | Yes |
| l2_leaf_reg | Num. | 0.5 to 30 | Yes |

3. We evaluate 500 HPCs when using random search and 250 HPC when using HEBO or SMAC3 (SMAC4HPO facade).

As resamplings, we use holdout with a 80/20 train-validation split and 5 folds for CV, so that the holdout strategy is just one fold of the CV and the fraction of data points being used for training and respectively validation are the same across different resampling strategies. 5-fold holdout simply repeats the holdout procedure five times and 5x 5-fold CV repeats the 5-fold CV five times. Each of the four resamplings can be reshuffled or not (standard).

As mentioned above, the test set is only varied for each of the 10 replica (repetitions with different seeds), but consistent for different tasks (i.e. the different learning algorithms are evaluated on the same test set, similarly, also the different dataset subsets all share the same test set). This allows for fair comparisons of different resamplings on a concrete problem (i.e. a given dataset, `train_valid` size and learning algorithm). Additionally, for the random search, the 500 HPCs evaluated for a given learning algorithm are also fixed over different dataset and `train_valid` size combinations. This is done to allow for an isolation of the effect, the concrete resampling (and whether it is reshuffled or not) has on generalization performance, reducing noise arising due to different HPCs. Learning algorithms themselves are not explicitly seeded to allow for variation during model training over different replications. Resamplings and partitioning of data are always performed in a stratified manner with respect to the target variable.

For the random search, we only ran (standard and reshuffled) holdout and (standard and reshuffled) 5x 5-fold CV experiments (because we can simulate 5-fold CV and 5-fold holdout experiments based

on the results obtained from the 5x 5-fold CV (by only considering the first repeat or the first fold for each of the five repeats).[9]

For running HEBO or SMAC3, each resampling (standard and reshuffled for holdout, 5-fold holdout, 5-fold CV, 5x 5-fold CV) has to be actually run due to the adaptive nature of BO.

For the random search experiments, this results in 10 (DGPs) $\times$ 3 (`train_valid` sizes) $\times$ 4 (learning algorithms) $\times$ 2 (holdout or 5x 5-fold CV) $\times$ 2 (standard or reshuffled) $\times$ 10 (replications) = 4800 HPO runs,[10] each involving the evaluation of 500 HPCs and each evaluation of an HPC involving either 2 (for holdout; due to retraining on train validation data) or 26 (for 5x 5-fold CV; due to retraining on train validation data) model fits. In summary, the random search experiments involve the evaluation of 2.4 Million HPCs with in total 33.6 Million model fits.

Similarly, for the HEBO and SMAC3 experiments, this each results in 10 (DGPs) $\times$ 3 (`train_valid` sizes) $\times$ 4 (learning algorithms) $\times$ 4 (holdout, 5-fold CV, 5x 5-fold CV or 5-fold holdout) $\times$ 2 (standard or reshuffled) $\times$ 10 (replications) = 9600 HPO runs[11], each involving the evaluation of 250 HPCs and each evaluation of an HPC involving either 2 (for holdout; due to retraining on train validation data), 6 (for 5-fold CV or 5-fold holdout; due to retraining on train validation data) or 26 (for 5x 5-fold CV; due to retraining on train validation data) model fits. In summary, the HEBO and SMAC3 experiments *each* involve the evaluation of 2.4 Million HPCs with in total 24 Million model fits.

### F.4   Compute Resources

We estimate our total compute time for the random search, HEBO and SMAC3 experiments to be roughly 11.86 CPU years. Benchmark experiments were run on an internal HPC cluster equipped with a mix of Intel Xeon E5-2670, Intel Xeon E5-2683 and Intel Xeon Gold 6330 instances. Jobs were scheduled to use a single CPU core and were allowed to use up to 16GB RAM. Total emissions are estimated to be an equivalent of roughly 6508.67 kg $CO_2$.

## G   Additional Benchmark Results Visualizations

### G.1   Main Experiments

In this section, we provide additional visualizations of the results of our benchmark experiments.

Figure 6 illustrates the trade-off between the final number of model fits required by different resamplings and the final average normalized test performance (AUC ROC) after running random search for a budget of 500 hyperparameter configurations. We can see that the reshuffled holdout on average comes close to the final test performance of the overall more expensive 5-fold CV.

Below, we give an overview of the different types of additional analyses and visualizations we provide. Normalized metrics, i.e., normalized validation or test performance refer to the measure being scaled to $[0, 1]$ based on the empirical observed minimum and maximum values obtained on the raw results level (ADTM; see Wistuba et al., 2018). More concretely, for each scenario consisting of a learning algorithm that is run on a given task (dataset $\times$ `train_valid` size) given a certain performance metric, the performance values (validation or test) for all resamplings and optimizers are normalized on the replication level to $[0, 1]$ by subtracting the empirical best value and dividing by the range of performance values. Therefore a normalized performance value of 0 is best and 1 is worst. Note that we additionally provide further aggregated results on the learning algorithm level and raw results of validation and test performance via https://github.com/slds-lmu/paper_2024_reshuffling.

- Random search
    - Normalized validation performance in Figure 7.

---

[9]We even could have simulated the vanilla holdout from the 5x 5-fold CV experiments by choosing an arbitrary fold and repeat but choose not to do so, to have some sanity checks regarding our implementation by being able to compare the "true" holdout with a the simulated holdout.

[10]Note that we do not have to take the 3 different metrics into account because random search allows us to simulate runs for different metric post hoc.

[11]Note that HEBO and SMAC3 were only run for ROC AUC as the performance metric.

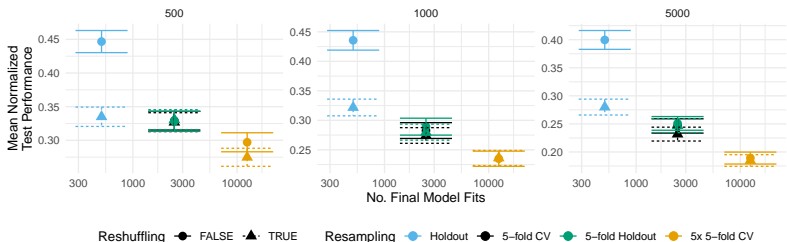

Figure 6: Trade-off between the final number of model fits required by different resamplings and the final average normalized test performance (AUC ROC) after running random search for a budget of $500$ hyperparameter configurations. Averaged over different tasks, learning algorithms and replications separately for increasing $n$ (train-validation sizes, columns). Shaded areas represent standard errors.

- – Normalized test performance in Figure 8.
- – Improvement in test performance over 5-fold CV in Figure 9.
- – Rank w.r.t. test performance in Figure 10.
- HEBO and SMAC3 vs. random search holdout
  - – Normalized validation performance in Figure 11.
  - – Normalized test performance in Figure 12.
  - – Improvement in test performance over standard holdout in Figure 13.
  - – Rank w.r.t. test performance in Figure 14.
- HEBO and SMAC3 vs. random search 5-fold holdout
  - – Normalized validation performance in Figure 15.
  - – Normalized test performance in Figure 16.
  - – Improvement in test performance over standard 5-fold holdout in Figure 17.
  - – Rank w.r.t. test performance in Figure 18.
- HEBO and SMAC3 vs. random search 5-fold CV
  - – Normalized validation performance in Figure 19.
  - – Normalized test performance in Figure 20.
  - – Improvement in test performance over 5-fold CV in Figure 21.
  - – Rank w.r.t. test performance in Figure 22.
- HEBO and SMAC3 vs. random search 5x 5-fold CV
  - – Normalized validation performance in Figure 23.
  - – Normalized test performance in Figure 24.
  - – Improvement in test performance over 5x 5-fold CV in Figure 25.
  - – Rank w.r.t. test performance in Figure 26.

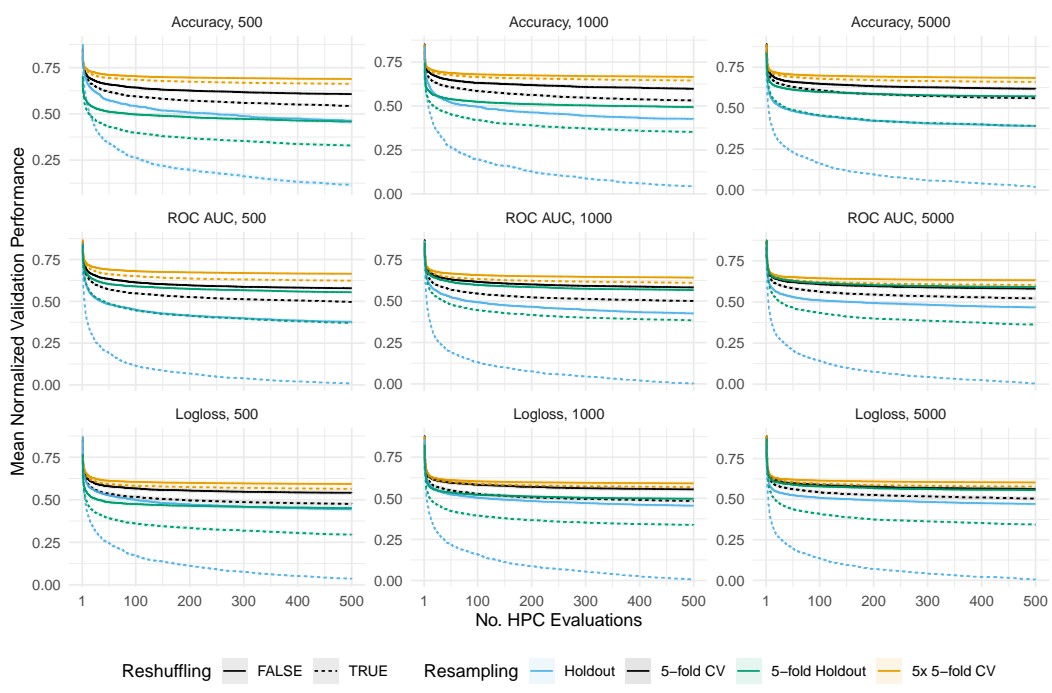

Figure 7: Random search. Average normalized performance over tasks, learners and replications for different $n$ (train-validation sizes, columns). Shaded areas represent standard errors.

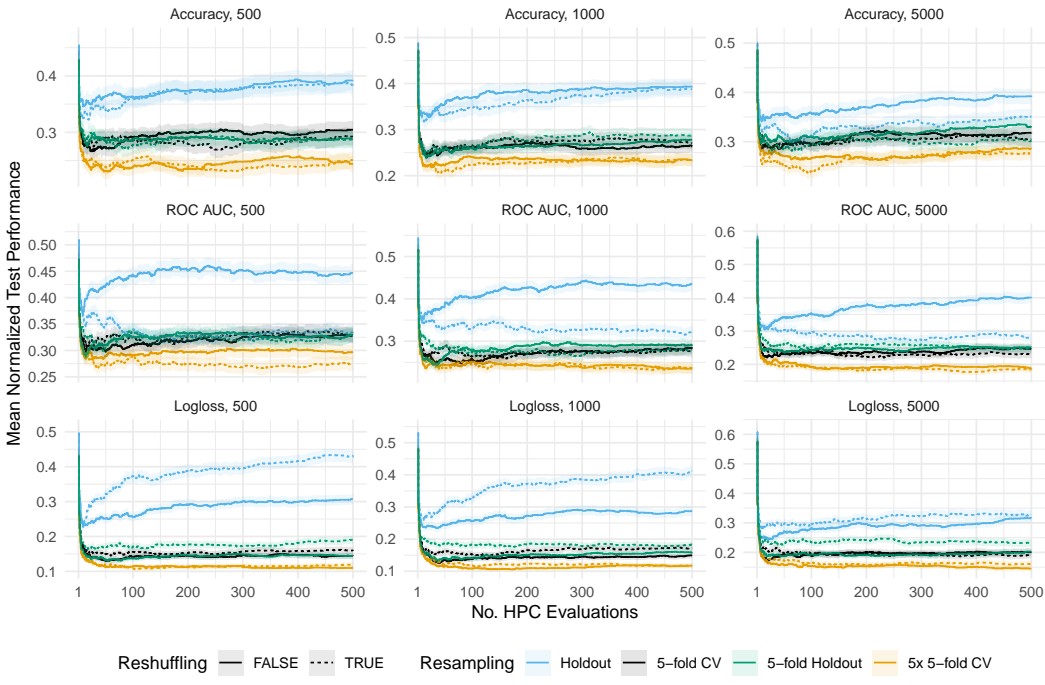

Figure 8: Random search. Average normalized test performance over tasks, learners and replications for different $n$ (train-validation sizes, columns). Shaded areas represent standard errors.

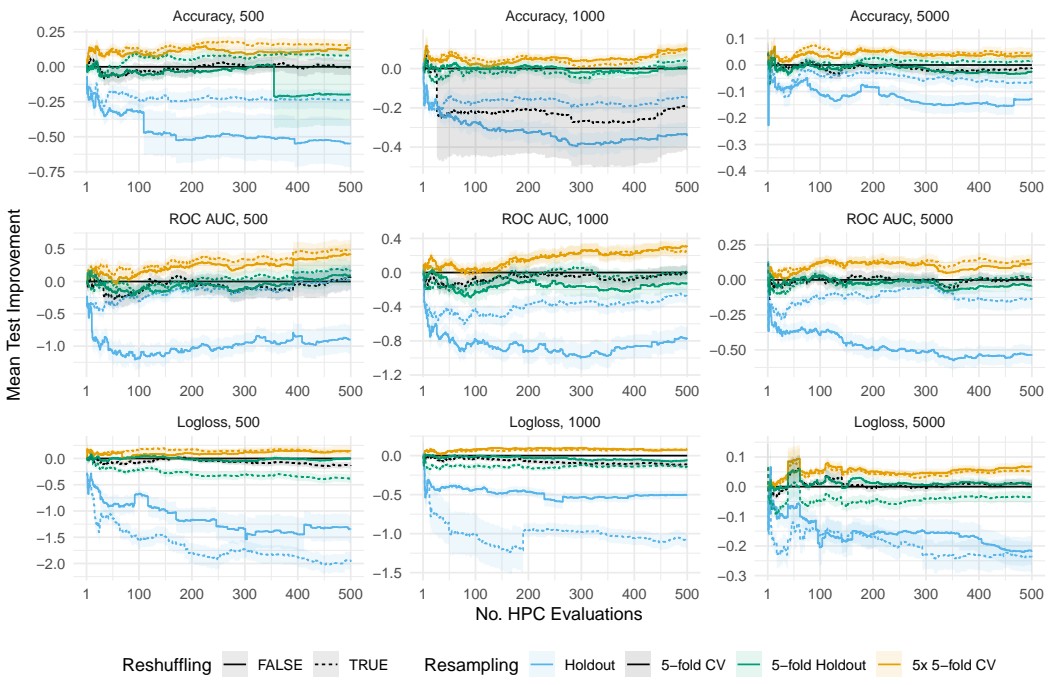

Figure 9: Random search. Average improvement (compared to standard 5-fold CV) with respect to test performance of the incumbent over tasks, learners and replications for different $n$ (train-validation sizes, columns). Shaded areas represent standard errors.

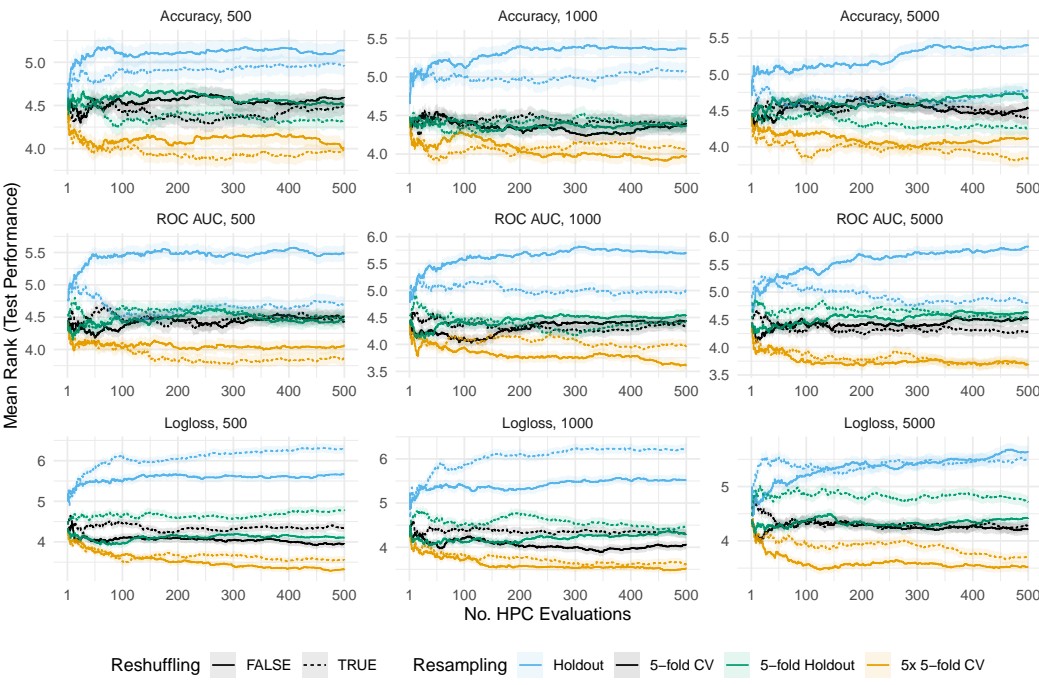

Figure 10: Random search. Average ranks (lower is better) with respect to test performance over tasks, learners and replications for different $n$ (train-validation sizes, columns). Shaded areas represent standard errors.

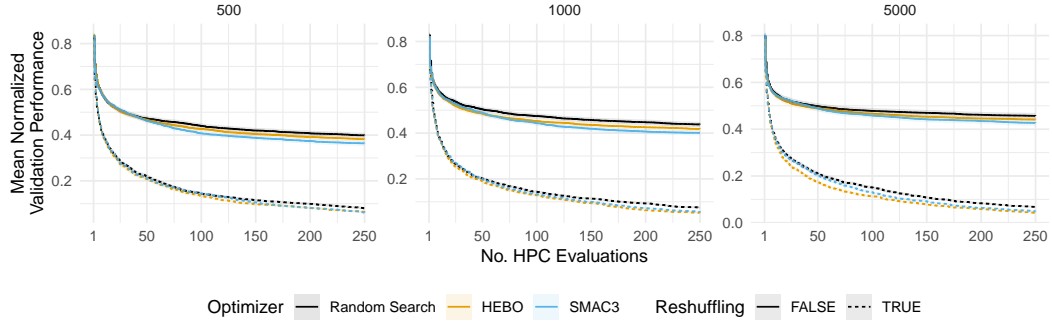

Figure 11: HEBO and SMAC3 vs. random search for holdout. Average normalized validation performance (ROC AUC) over tasks, learners and replications for different $n$ (train-validation sizes, columns). Shaded areas represent standard errors.

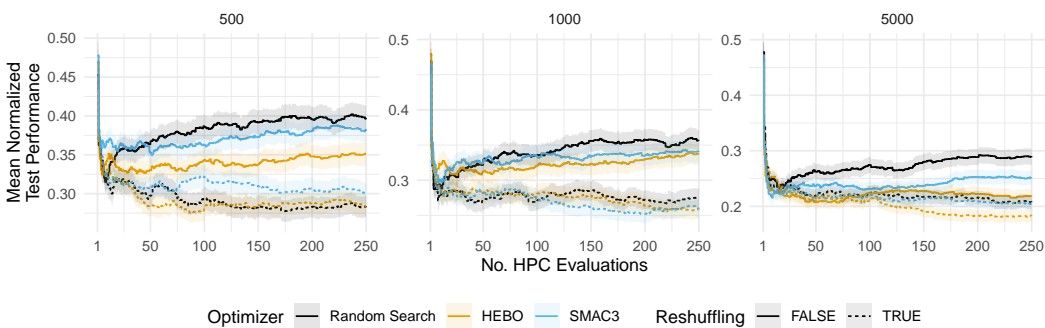

Figure 12: HEBO and SMAC3 vs. random search for holdout. Average normalized test performance (ROC AUC) over tasks, learners and replications for different $n$ (train-validation sizes, columns). Shaded areas represent standard errors.

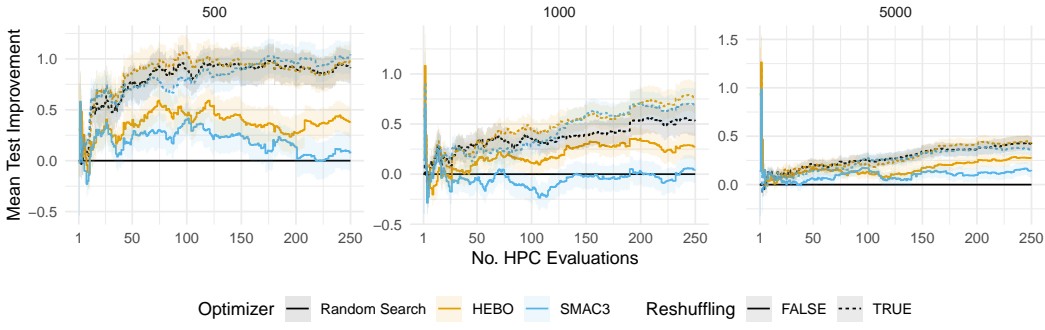

Figure 13: HEBO and SMAC3 vs. random search for holdout. Average improvement (compared to standard holdout) with respect to test performance (ROC AUC) of the incumbent over tasks, learners and replications for different $n$ (train-validation sizes, columns). Shaded areas represent standard errors.

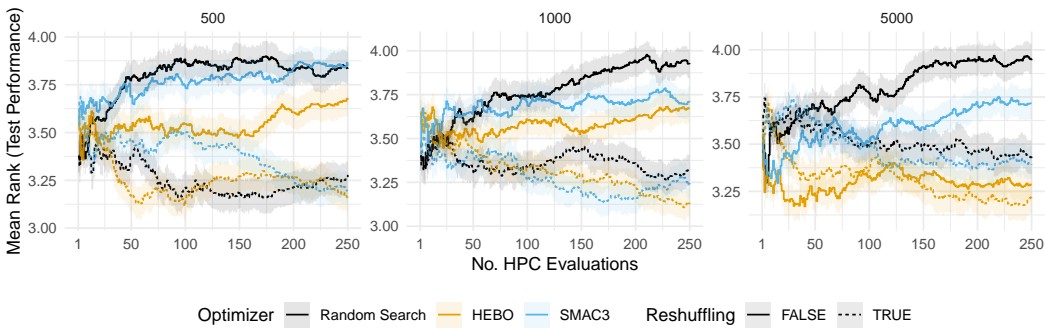

Figure 14: HEBO and SMAC3 vs. random search for holdout. Average ranks (lower is better) with respect to test performance (ROC AUC) of the incumbent over tasks, learners and replications for different $n$ (train-validation sizes, columns). Shaded areas represent standard errors.

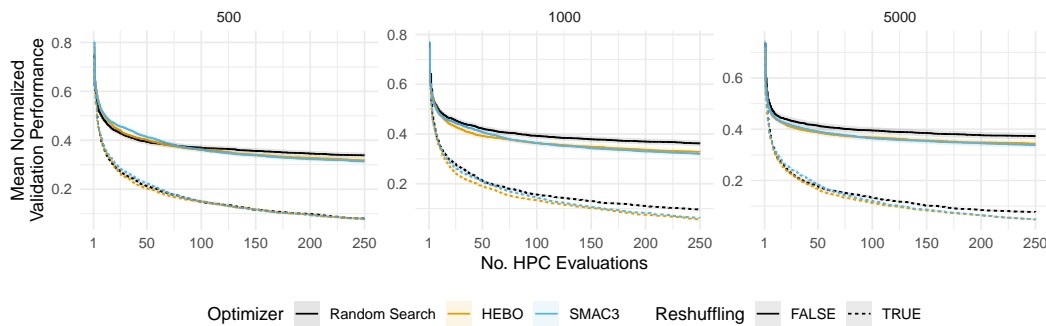

Figure 15: HEBO and SMAC3 vs. random search for 5-fold holdout. Average normalized validation performance (ROC AUC) over tasks, learners and replications for different $n$ (train-validation sizes, columns). Shaded areas represent standard errors.

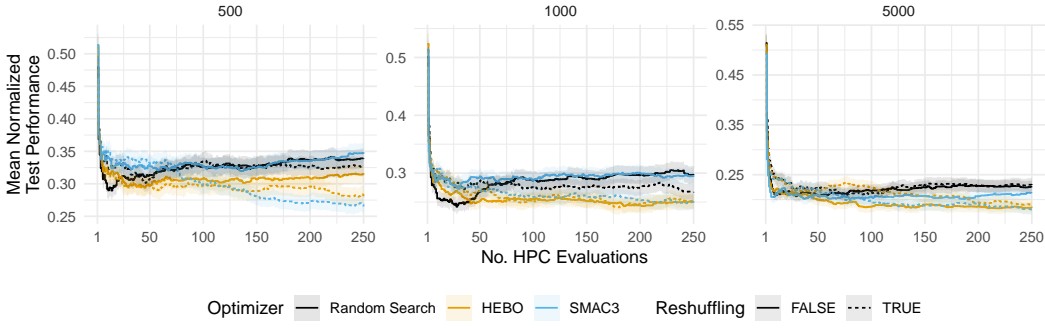

Figure 16: HEBO and SMAC3 vs. random search for 5-fold holdout. Average normalized test performance (ROC AUC) over tasks, learners and replications for different $n$ (train-validation sizes, columns). Shaded areas represent standard errors.

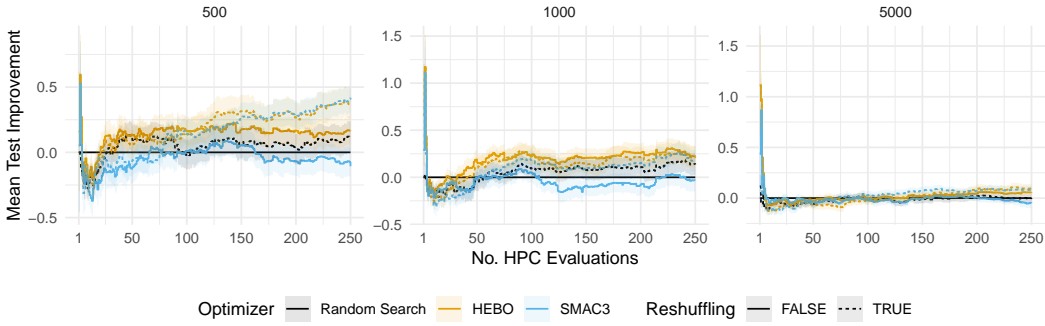

Figure 17: HEBO and SMAC3 vs. random search for 5-fold holdout. Average improvement (compared to standard 5-fold holdout) with respect to test performance (ROC AUC) of the incumbent over tasks, learners and replications for different $n$ (train-validation sizes, columns). Shaded areas represent standard errors.

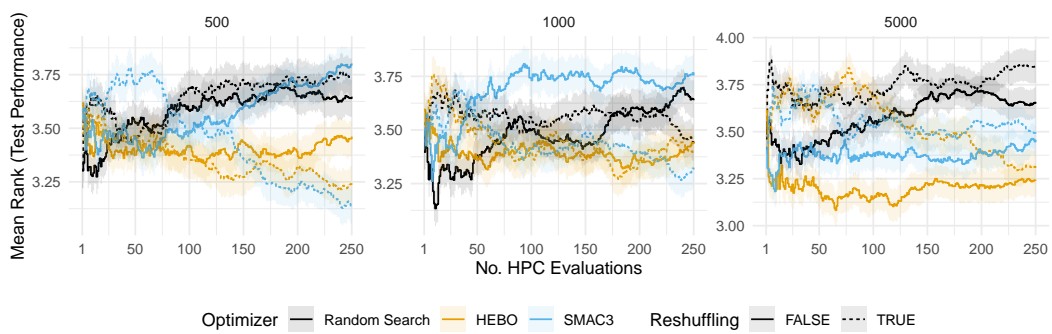

Figure 18: HEBO and SMAC3 vs. random search for 5-fold holdout. Average ranks (lower is better) with respect to test performance (ROC AUC) of the incumbent tasks, learners and replications for different $n$ (train-validation sizes, columns). Shaded areas represent standard errors.

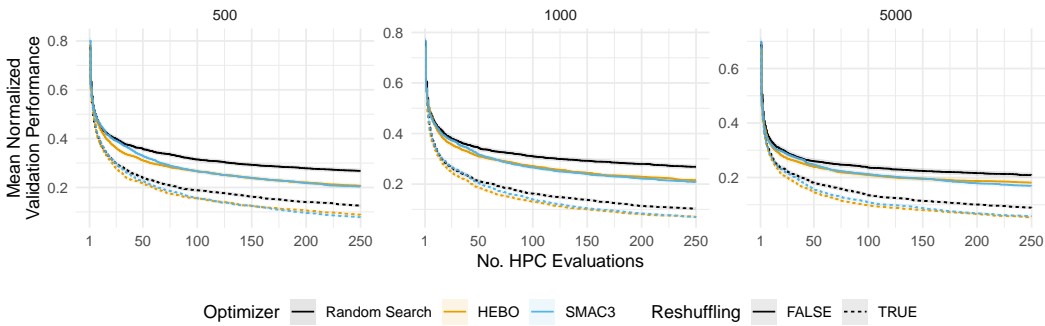

Figure 19: HEBO and SMAC3 vs. random search for 5-fold CV. Average normalized validation performance (ROC AUC) over tasks, learners and replications for different $n$ (train-validation sizes, columns). Shaded areas represent standard errors.

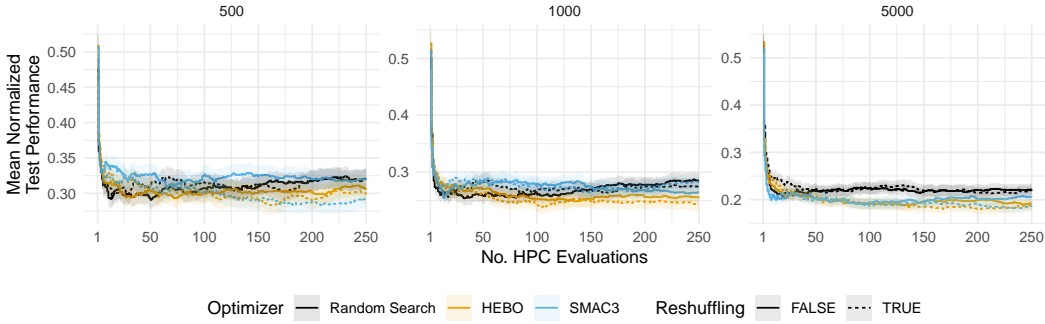

Figure 20: HEBO and SMAC3 vs. random search for 5-fold CV. Average normalized test performance (ROC AUC) over tasks, learners and replications for different $n$ (train-validation sizes, columns). Shaded areas represent standard errors.

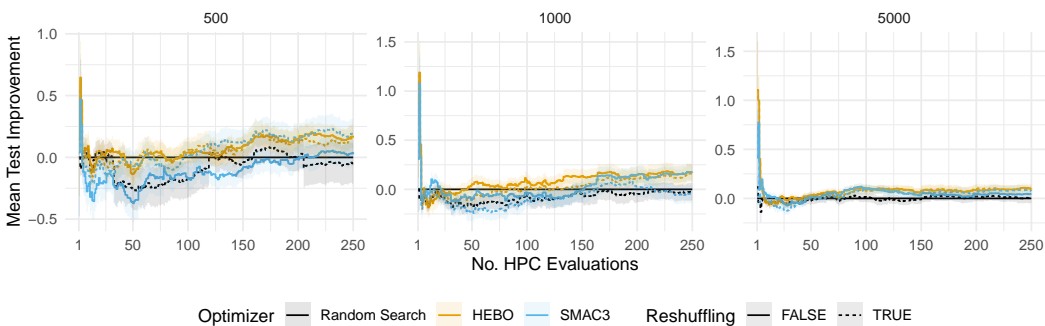

Figure 21: HEBO and SMAC3 vs. random search for 5-fold CV. Average improvement (compared to standard 5-fold CV) with respect to test performance (ROC AUC) of the incumbent over tasks, learners and replications for different $n$ (train-validation sizes, columns). Shaded areas represent standard errors.

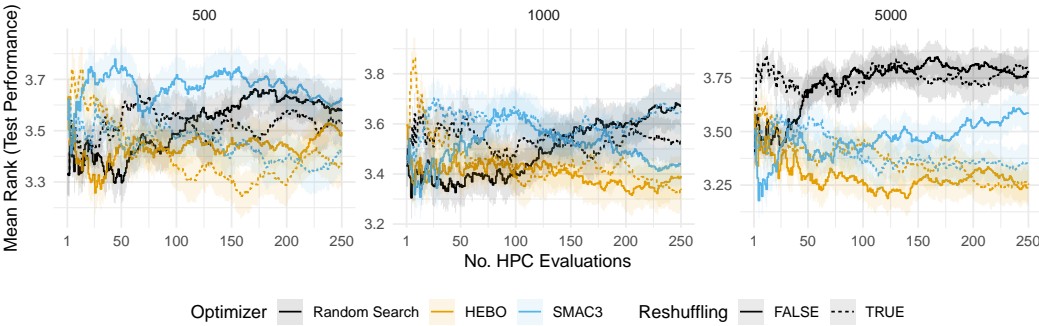

Figure 22: HEBO and SMAC3 vs. random search for 5-fold CV. Average ranks (lower is better) with respect to test performance (ROC AUC) of the incumbent over tasks, learners and replications for different $n$ (train-validation sizes, columns). Shaded areas represent standard errors.

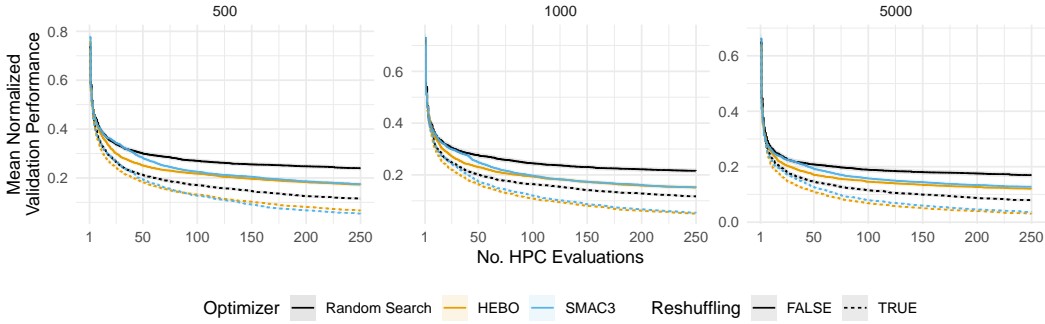

Figure 23: HEBO and SMAC3 vs. random search for 5x 5-fold CV. Average normalized validation performance (ROC AUC) over tasks, learners and replications for different $n$ (train-validation sizes, columns). Shaded areas represent standard errors.

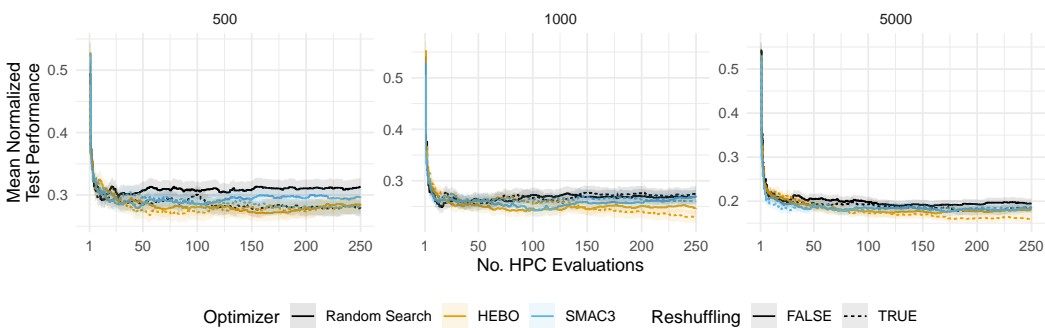

Figure 24: HEBO and SMAC3 vs. random search for 5x 5-fold CV. Average normalized test performance (ROC AUC) over tasks, learners and replications for different $n$ (train-validation sizes, columns). Shaded areas represent standard errors.

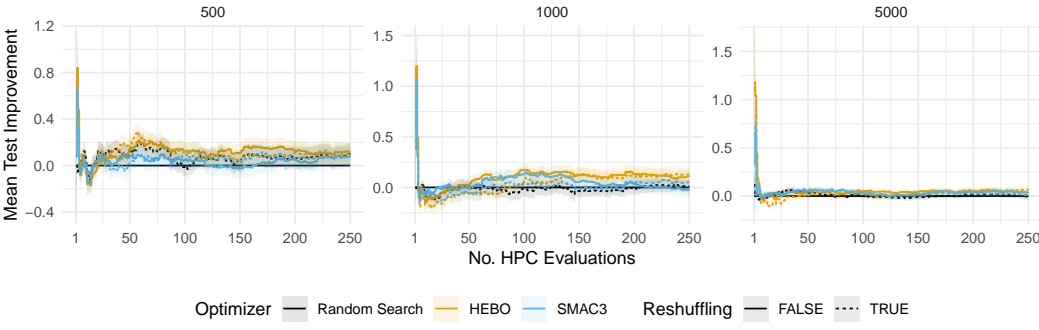

Figure 25: HEBO and SMAC3 vs. random search for 5x 5-fold CV. Average improvement (compared to standard 5x 5-fold CV) with respect to test performance (ROC AUC) of the incumbent over tasks, learners and replications for different $n$ (train-validation sizes, columns). Shaded areas represent standard errors.

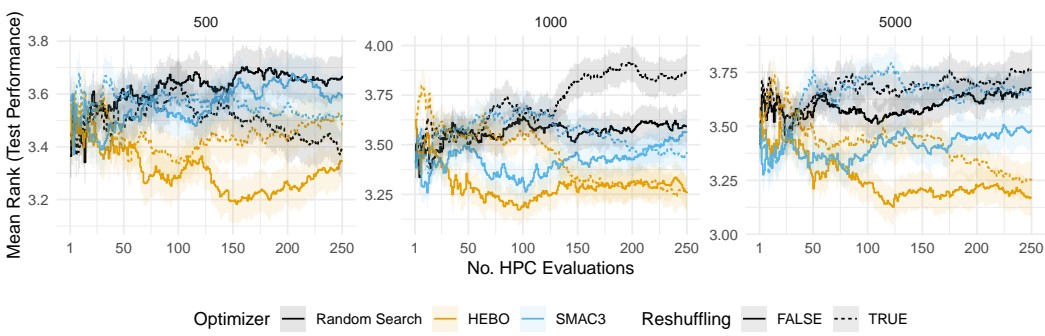

Figure 26: HEBO and SMAC3 vs. random search for 5x 5-fold CV. Average ranks (lower is better) with respect to test performance (ROC AUC) of the incumbent over tasks, learners and replications for different $n$ (train-validation sizes, columns). Shaded areas represent standard errors.

### G.2 Ablation on M-fold holdout

Based on the 5x 5-fold CV results we further simulated different $M$-fold holdout resamplings (standard and reshuffled) by taking M repeats from the first fold of the 5x 5-fold CV. This allows us to get an understanding of the effect more folds have on $M$-fold holdout, especially in the context of reshuffling.

Regarding normalized validation performance we observe that more folds generally result in a less optimistically biased validation performance (see Figure 27). Looking at normalized test performance (Figure 28) we observe the general trend that more folds result in better test performance – which is expected. Reshuffling generally results in better test performance compared to the standard resampling (with the exception of logloss where especially in the case of a single holdout, reshuffling can hurt generalization performance). This effect is smaller, the more folds are used, which is in line with our theoretical results presented in Table 1. Looking at improvement compared to standard 5-fold holdout with respect to test performance and ranks with respect to test performance, we observe that often reshuffled 2-fold holdout results that are highly competitive with standard 3, 4 or 5-fold holdout.

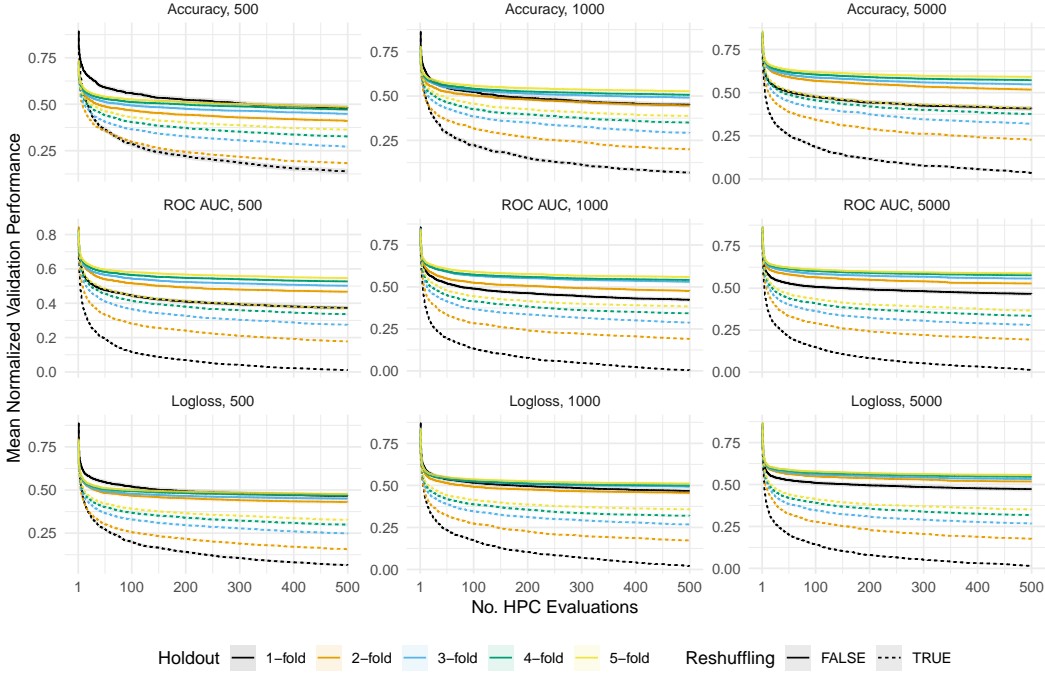

Figure 27: Random search. Average normalized validation performance over tasks, learners and replications for different $n$ (train-validation sizes, columns). Shaded areas represent standard errors.

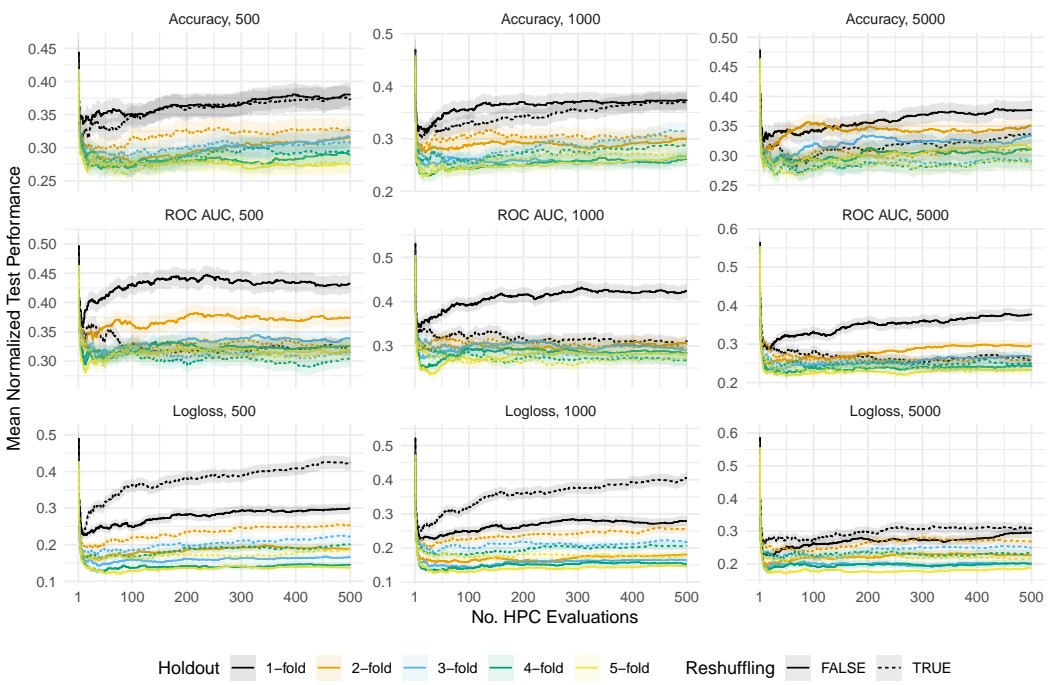

Figure 28: Random search. Average normalized test performance over tasks, learners and replications for different $n$ (train-validation sizes, columns). Shaded areas represent standard errors.

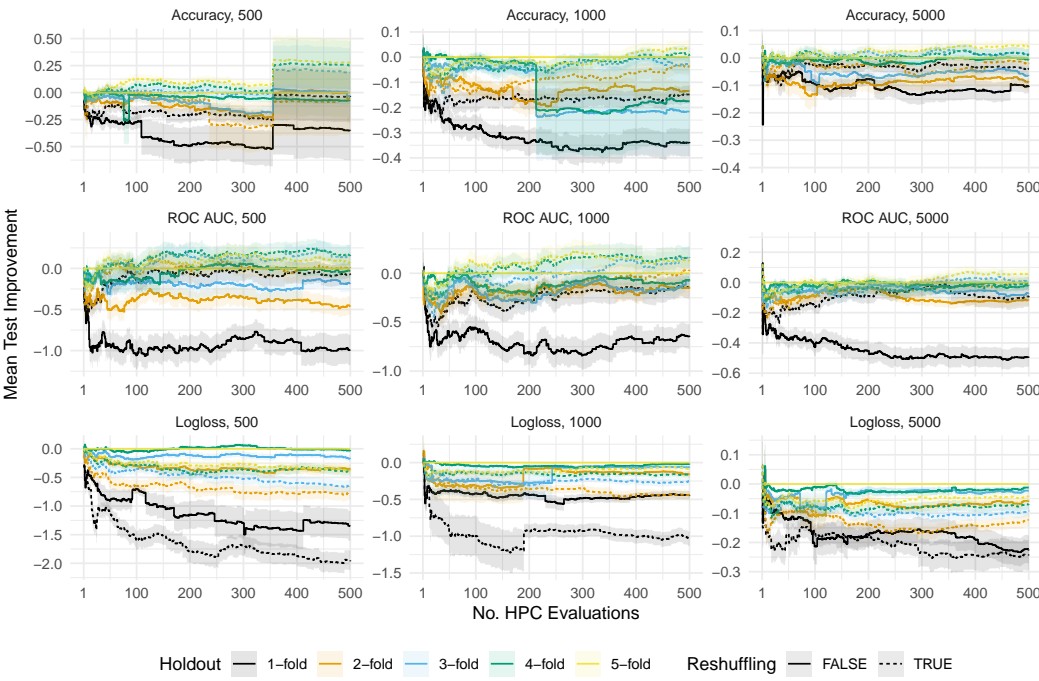

Figure 29: Random search. Average improvement (compared to standard 5-fold holdout) with respect to test performance of the incumbent over tasks, learners and replications for different $n$ (train-validation sizes, columns). Shaded areas represent standard errors.

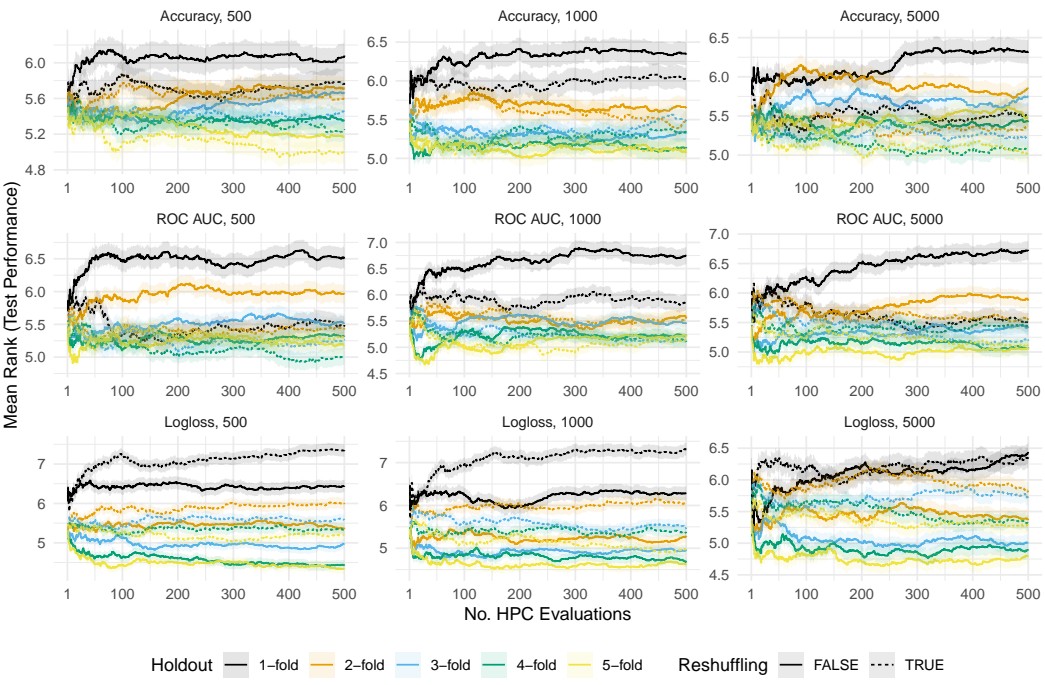

Figure 30: Random search. Average ranks (lower is better) with respect to test performance of the incumbent over tasks, learners and replications for different $n$ (train-validation sizes, columns). Shaded areas represent standard errors.

