# OpenReview forum: "Reshuffling Resampling Splits Can Improve Generalization of Hyperparameter Optimization"
_NeurIPS.cc/2024/Conference — NeurIPS 2024 poster_

### Official Review · Reviewer_rcFG · 2024-07-07

**Soundness:** 3
**Presentation:** 4
**Contribution:** 3
**Rating:** 7
**Confidence:** 3

**Summary:**

The paper presents a compelling investigation into the effects of reshuffling resampling splits on the generalization performance of hyperparameter optimization (HPO) strategies. Through a mix of theoretical analysis and empirical studies, the authors argue that reshuffling can lead to statistically significant improvements in model performance, especially when the loss surface is flat and its estimate is noisy.

**Strengths:**

Novel Approach: The paper introduces a new approach to HPO by proposing the reshuffling of train-validation splits. This contrasts with the fixed splits to ensure consistency across evaluations.

Theoretical and Empirical Validation: The combination of theoretical insights and empirical validation is a strong aspect of this paper. The theoretical model explaining how reshuffling affects the asymptotic behavior of the validation loss surface is particularly insightful.

Practical Implications: The findings suggests that reshuffling could be a simple yet effective technique to improve the robustness of hyperparameter tuning.

**Weaknesses:**

Generalizability of Results: The experiments are somewhat limited in scope, focusing on specific types of data sets and HPO strategies. It would be beneficial to see broader testing across more varied data types and with different models to fully understand the generalizability of the results.

**Questions:**

There is little discussion on the computational costs associated with reshuffling, which could be significant, especially in large-scale applications. Understanding this trade-off is crucial for practical adoption.

**Limitations:**

As mentioned in the paper.

---

> ### Author Rebuttal · Authors · 2024-08-06
>
> Thank you very much for your positive review of our submission and your suggestions on how to improve our submission. Below you can find the answers to your criticism and questions:
>
> > Generalizability of Results: The experiments are somewhat limited in scope, focusing on specific types of data sets and HPO strategies. It would be beneficial to see broader testing across more varied data types and with different models to fully understand the generalizability of the results.
>
> Please see our general response, point 1.
>
> > There is little discussion on the computational costs associated with reshuffling, which could be significant, especially in large-scale applications. Understanding this trade-off is crucial for practical adoption.
>
> Shuffling the data amounts to drawing a random permutation of the observation indices, which comes at almost no computational cost. Shuffling takes less than a millisecond, which is negligible compared to the costs of model fits and/or the overhead of BO. We will add a comment to the paper.

---

### Official Review · Reviewer_QU3x · 2024-07-10

**Soundness:** 3
**Presentation:** 3
**Contribution:** 3
**Rating:** 6
**Confidence:** 4

**Summary:**

The standard protocol evaluates various hyperparameter configurations using a resampling estimate of the generalization error to guide optimization and select a final hyperparameter configuration. The authors argue that reshuffling the splits for every configuration often improves the final model’s generalization performance on unseen data. They provide a theoretical analysis explaining how reshuffling affects the asymptotic behavior of the validation loss surface and provides a bound on the expected regret in the limiting regime.

The paper also presents a simulation study confirming the theoretical results and demonstrates the practical usefulness of reshuffling in a large-scale, realistic hyperparameter optimization experiment. The authors found that reshuffling leads to test performances that are competitive with using fixed splits, and it drastically improves results for a single train-validation holdout protocol.

**Strengths:**

The paper provides a rigorous theoretical analysis explaining how reshuffling the splits affects the asymptotic behavior of the validation loss surface. This analysis helps establish the foundation for understanding the benefits of reshuffling.

The authors conduct a simulation study to validate their theoretical results. By demonstrating that reshuffling leads to competitive test performances and improved results in a realistic hyperparameter optimization experiment, they provide empirical evidence supporting their claims.

The technique of reshuffling resampling splits is directly applicable to hyperparameter optimization in machine learning. It addresses a common challenge faced by practitioners and provides a practical solution to improve model generalization.

**Weaknesses:**

The effectiveness of reshuffling depends on dataset characteristics, such as its size, noise level, and correlation structure, which limits the usefulness of reshuffling.

**Questions:**

I am interested in seeing how reshuffling performs on more complex benchmarks and with more sophisticated models, such as training Resnet on CIFAR-10/100 or Imagenet. Experiments in such realistic settings could further strengthen the results.

**Limitations:**

The authors acknowledge that the benefits of reshuffling may vary based on noise correlation and loss surface curvature. I believe that the authors have conducted enough discussion regarding the limitations of the paper

---

> ### Author Rebuttal · Authors · 2024-08-06
>
> Thank you very much for your positive review of our submission and for assessing our study as “large-scale, realistic hyperparameter optimization experiment”. Below you can find detailed answers to your questions and criticism:
>
> > The effectiveness of reshuffling depends on dataset characteristics, such as its size, noise level, and correlation structure, which limits the usefulness of reshuffling.
>
> While we fully agree with the statement that the effectiveness of reshuffling depends on the characteristics of the optimization problem (we assume that this is what you mean by “dataset characteristics”), we do not think this is a weakness. On the contrary, a significant contribution of our paper is to establish, for the first time, an explicit connection between generalization and dataset characteristics. We are confident this will prove useful in many applications and future methodological developments.
>
> > I am interested in seeing how reshuffling performs on more complex benchmarks and with more sophisticated models, such as training Resnet on CIFAR-10/100 or Imagenet. Experiments in such realistic settings could further strengthen the results.
>
> Please see our general response, point 1.

---

> > ### Comment · Reviewer_QU3x · 2024-08-12
> >
> > Thank you for addressing my comments. I will keep my current score.

---

### Official Review · Reviewer_8FSs · 2024-07-10

**Soundness:** 3
**Presentation:** 3
**Contribution:** 3
**Rating:** 6
**Confidence:** 3

**Summary:**

This paper studies the effect of reshuffling the splits over which hyper-parameter optimiazation is performed. Specifically, the authors provide theoretical guarantees on the generalization performance achieved via reshuffling and empirically demonstrate its impact both via simulation and benchmark experiments. Overall, the results demonstrate that reshuffling improves HPO performance according to the theoretical analysis.

**Strengths:**

**Originality:**
Related work seems to be adequately cited and, to the best of my knowledge, this work explores a dimension of the hyper-parameter tuning problem that I had not seen explored yet.

&NewLine;
**Quality:**

Overall, the paper is well written and easy to follow. It is also generally well organised and self-contained. I did not carefully check the mathematical results and deductions. I particularly appreciated the discussion on the limitations of the study.

&NewLine;
**Clarity:**

[Line 94] You introduce some initial observations for different selection methods, but you haven't yet introduced the methods. There seems to be too big of a jump from the equations to the observations for the several methods. Similarly, it would have helped me better understand the reshuffling idea if Table 1 had more explanation. First, the meaning of "reshuffling effect" is not clear at this stage of the paper. Second, I think it would help the reader if you clarified whether increases/decreases in tau and sigma are desirable and why.

There are a lot of variables introduced throughout (which is absolutely normal) but this also means the reader may forget what some of them mean as they are reading and new variables are introduced. It could be helpful to periodically remind the reader of what some variables mean (especially if they have been introduced some paragraphs ago). (For example, as you do in lines 215-216).

Some things seem to come out of the blue: e.g. line 140 -- "there are two regimes". Did you figure this out while deriving the equations or did you suspect this a priori? Where do these two regimes come from?

Just like you illustrate one of the scenarios via Figure 1, you could have a similar figure illustrating the case when the process is correlated.

It would be quite useful for a reader if you had some form of pseudo-code highlighting the reshuffling procedure.


&NewLine;
**Significance:**

Although not explicitly highlighted in the paper, it seems to me that the idea of reshuffling is interesting as it can contribute significantly to finding better configurations at a cheaper cost.

**Weaknesses:**

See above for weaknesses

**Questions:**

**a)**
How do you derive Equation (1)? Are the observations that follow based only off of theorem 2.1 and Equation (1)? Or based on other equations and theorems?

&NewLine;
**Comments:**

You should repeat (on the figures) the meaning of 500, 1000, and 5000. It would make it much easier to read/interpret the results.

It seems that if you were to plot something like mean test improvement as a function of the cost  of each HPO approach, the benefits of shuffling would be more visible. From the current plots, it is not clear that the advantage of trying to improve the performance of holdout (especially when there are better approaches, as you show) is that holdout is cheaper than these other (better performing) approaches.

&NewLine;
**[minor comment / observation]**

[line 26] You should explicitly introduce the CV acronym

[line 204] repeated 'sized'

**Limitations:**

The authors adequately discussed the limitations of the work.

---

> ### Author Rebuttal · Authors · 2024-08-06
>
> Thank you very much for your positive review and the suggestions, which will help us to improve our paper. We are very happy that you find our paper well-written and that you appreciate our up-front discussion of the limitations of the study. Below you can find detailed answers to your questions.
>
> > [Line 94] You introduce some initial observations for different selection methods, but you haven't yet introduced the methods. There seems to be too big of a jump from the equations to the observations for the several methods. Similarly, it would have helped me better understand the reshuffling idea if Table 1 had more explanation. First, the meaning of "reshuffling effect" is not clear at this stage of the paper. Second, I think it would help the reader if you clarified whether increases/decreases in tau and sigma are desirable and why. [...] a) How do you derive Equation (1)? Are the observations that follow based only off of theorem 2.1 and Equation (1)? Or based on other equations and theorems?
>
> Thank you for raising this issue and for the great suggestions. Please refer to our general comment, point 2, for some clarification. By “reshuffling effect” we simply mean “the difference in behavior between the unshuffled and shuffled variant of a resampling method”. We will add more explanation for Table 1 and discuss whether and why increase/decrease in $\tau$ and $\sigma$ are desirable as suggested.
>
> > There are a lot of variables introduced throughout (which is absolutely normal) but this also means the reader may forget what some of them mean as they are reading and new variables are introduced. It could be helpful to periodically remind the reader of what some variables mean (especially if they have been introduced some paragraphs ago). (For example, as you do in lines 215-216).
>
> We have revised the papers and added some reminders as suggested. Further, as suggested by another reviewer, we now include a Table of symbols and their meaning in the appendix (a preview is also shown in the attached PDF).
>
> > Some things seem to come out of the blue: e.g. line 140 -- "there are two regimes". Did you figure this out while deriving the equations or did you suspect this a priori? Where do these two regimes come from?
>
> The two regimes became apparent after our derivations. When looking at the final bound, reshuffling can only help when the term $A(\tau)$ is strictly positive. This is exactly the case when $\sigma / 2m \eta^2 \le e$. Intuitively speaking, the two regimes come from the fact that reshuffling always adds some noise to the HPO process. Our analysis shows that this can be helpful, but only if the HPO problem is “sufficiently hard” in the first place. If $\sigma / 2m \eta^2 \le e$ the curvature is very high compared to the noise, so it is extremely easy to find the best configuration. We will revise our discussion of the regimes accordingly.
>
> > Just like you illustrate one of the scenarios via Figure 1, you could have a similar figure illustrating the case when the process is correlated.
>
> Thanks for the suggestion. We added a version of Figure 1 where the correlation in the (unshuffled) empirical loss is so high that reshuffling doesn’t help, and refer to it in the main text. For your convenience, the figure is also provided in the PDF accompanying the rebuttal.
>
> > It would be quite useful for a reader if you had some form of pseudo-code highlighting the reshuffling procedure.
>
> Thank you very much for your suggestion, but we find it difficult to comply. The method is so simple (1. Randomly permute the observation indices, 2. Do resampling as usual) that it would feel odd to write this in pseudo-code. If you think this would benefit the paper or we misunderstood the request, we are of course happy to reconsider.
>
> > You should repeat (on the figures) the meaning of 500, 1000, and 5000. It would make it much easier to read/interpret the results.
>
> Good idea; we will update the figures accordingly.
>
> > It seems that if you were to plot something like mean test improvement as a function of the cost of each HPO approach, the benefits of shuffling would be more visible. From the current plots, it is not clear that the advantage of trying to improve the performance of holdout (especially when there are better approaches, as you show) is that holdout is cheaper than these other (better performing) approaches.
>
> We have added a plot in the newly provided PDF which shows the trade-off between the final number of model fits required by different resamplings and the final test performance. We can see that the reshuffled holdout on average comes close to the final test performance of the more expensive 5-fold CV. A similar plot be added to the final version of the paper.
>
> > [minor comment / observation]
>
> Thank you very much for finding these two mistakes, we will of course correct them.

---

> > ### Comment · Reviewer_8FSs · 2024-08-11
> >
> > Thank you for the additional plots and for the clarifications.
> >
> > **Pseudo-code:** I understand the method is quite simple, but the theoretical study in the paper is quite complex, making something simple seem rather (and perhaps unnecessarily) complex. This can arguably be seen as a disadvantage, as simplicity is typically something hard to attain.
> >
> > **Practical consideration/usage:** although I see advantages in using resampling, from a users' point of view, if I wanted to benefit from reshuffling I would have to try and compare different methods to evaluate, empirically, which one worked best for my use-case/dataset in practice. This seems rather burdensome and impractical.

---

> ### Author Response · Authors · 2024-08-11
>
> PseudoCode: We can surely add it, if you think it improves readability, no problem.
>
> Practical consideration/usage: We disagree here, please reconsider this:
> Especially for holdout, there is a clear on average advantage using reshuffling. This is clearly demonstrated by our experiments. Staying with the current "state" / "recommendation"  (no reshuffling) means you will perform worse, in many experiments. Just because we provide a deeper theoretical analysis, which shows in which situations this is most beneficial, and because we currently cannot exactly pinpoint them in a practical manner, shouldn't have us ignore this piece of knowledge now. We think your criticism would be merited if the situations where resampling / HPO are improved are somewhat "specific and rare", and then we would not help the reader to identify these situations. But as explained above (and the paper) this is not the case.

---

> > ### Comment · Reviewer_8FSs · 2024-08-13
> >
> > I'm not pushing to have the pseudo-code in. But I liked your simplistic formulation of the problem. I think that simplified view is a nice add-on.  Especially before going into the theoretical demonstrations or throughout the text to break away from the mathematical equations and notation. Sentences that describe the tasks/problems in a simplified away provide good break points for readers to absorb/digest all the information they've read until that point.
> >
> > I see what you mean for holdout. I still believe that if the HPO problem is not defined on binary classification, I might have to test and compare multiple approaches. I don't think this demerits your current work/contribution. But I believe this is something that can be improved/studied in the future.

---

### Official Review · Reviewer_sYf9 · 2024-07-11

**Soundness:** 2
**Presentation:** 2
**Contribution:** 2
**Rating:** 4
**Confidence:** 3

**Summary:**

In hyperparameter optimization (HPO), individuals often use the same resampling for different configurations to ensure fair comparison. However, this fixed split may introduce bias into the optimization process, particularly after numerous evaluations, as it tends to favor configurations that align well with that specific split. This study systematically explores the effects of reshuffling on HPO performance through theoretical analysis, controlled simulation studies, and real dataset experiments. The authors highlight that reshuffling resampling splits during HPO can help find configurations with superior overall generalization, particularly in scenarios where the loss surface is flat and estimates are noisy. Experimental findings indicate that when holdout reshuffling is employed, the ultimate generalization performance often rivals that of 5-fold cross-validation in a broad array of settings.

**Strengths:**

1. This paper is well organized, with rich mathematical expressions and clear experimental figures.
2. The authors delve deeply into the effect of reshuffling in HPO from theoretical analysis, simulation studies, and experiments on realistic datasets, highlighting its significance especially when the loss surface is float and estimates are noisy.
3. In the experiments, the authors study the effect of reshuffling with different resampling methods and HPO methods. The design and visual presentation of the experiments is clear.

**Weaknesses:**

1. The authors use a large number of symbols in the paper, adding a notation table may be considered to enhance readability.
2. The authors primarily investigate the impact of reshuffling in resampling and HPO but do not propose novel methods. The actual implementation is simple, with $n\in\{500,1000,5000\}$. The authors could consider making improvements to reshuffling itself based on the analysis of dataset noise and surface to determine reshuffling strategies.
3. The paper could consider conducting experiments on a wider range of models (e.g., SVM), datasets (e.g., mnist and cifar10), and HPO methods (e.g., successive halving) to demonstrate the effects of reshuffling comprehensively.

**Questions:**

1. In practical applications, reshuffling implies an increase in time consumption. How do the different methods in the experiments of this paper perform in terms of time consumption? When the total time is constant, does using reshuffling have advantages over using only resampling, or what advantages does it have?
2. In the experiments of this paper, how do the methods perform on each dataset? Does reshuffling exhibit different performances on different datasets?
3. Successive halving, as a method that uses different sampling ratios in the optimization process, how does reshuffling perform in it?

**Limitations:**

Already discussed.

---

> ### Author Rebuttal · Authors · 2024-08-06
>
> Thank you very much for your suggestions on the presentation and the experimental setup, which will help us make our paper more convincing. In the following, we provide detailed answers to your questions:
>
> >The authors use a large number of symbols in the paper, adding a notation table may be considered to enhance readability.
>
> This is a very good idea. We will include this in the appendix of our paper, and we already show a preview in the attached PDF.
>
> > The authors primarily investigate the impact of reshuffling in resampling and HPO but do not propose novel methods. The actual implementation is simple, with n∈500,1000,5000. The authors could consider making improvements to
> reshuffling itself based on the analysis of dataset noise and surface to determine reshuffling strategies.
>
> It is true that we primarily investigate the impact of reshuffling in HPO. We firmly believe that providing theoretical and empirical insights into an arguably important method/field like HPO is extremely valuable in itself (and traditionally valued at venues like NeurIPS). Further, the technique is immediately available, and we think that our quite large-scale experiments convincingly demonstrate its effectiveness (see also the reviews of rcFG and QU3x, which seem to strongly agree with our argument here). Building more sophisticated shuffling methods exploiting data set characteristics is, however, beyond the foundational scope of our paper.
>
> > The paper could consider conducting experiments on a wider range of models (e.g., SVM), datasets (e.g., mnist and cifar10), and HPO methods (e.g., successive halving) to demonstrate the effects of reshuffling comprehensively.
>
> Please see our general response, point 1.
>
> > In practical applications, reshuffling implies an increase in time consumption. How do the different methods in the experiments of this paper perform in terms of time consumption? When the total time is constant, does using reshuffling have advantages over using only resampling, or what advantages does it have?
>
> We think this is a slight misunderstanding. Shuffling the data amounts to drawing a random permutation of the observation indices and comes at pretty much no additional computational cost. Shuffling takes less than a millisecond, which is negligible compared to the costs of model fits / the overhead of BO. The advantage of the approach is better generalization performance in many cases, and you basically get this "free of charge", which we think is indeed a nice result. Of course, there is no “free lunch”, which is explained by our theoretical analysis. But as our experiments clearly show: On average you often run better in HPO with reshuffling than without. We will state this more clearly in the paper.
>
> > In the experiments of this paper, how do the methods perform on each dataset? Does reshuffling exhibit different performances on different datasets?
>
> Experimental results per data set are already available in our anonymous repository (link in Appendix F; navigate to e.g., plots/catboost_accuracy/test.pdf). Reshuffling can exhibit different performances on different datasets but results in a strong performance on average, as exemplified in the aggregated plots in the main paper and Appendix.
>
> > Successive halving, as a method that uses different sampling ratios in the optimization process, how does reshuffling perform in it?
>
> While we think that this could later be an interesting direction to explore, we think this is out-of-scope for the current paper. SH uses subsampling of the data to address the multi-fidelity problem in HPO (and SH is not restricted to subsampling, other ways exist to reduce compute cost, such as lower numbers of epochs). We already evaluate the method in complex contexts (data sets, learners, complex HPO methods), and adding a further (complex) aspect doesn't seem helpful in getting to the "core effect" of reshuffling.
>
> But please note that we instead opted to add further experiments on a different SOTA HPO method; see general comment 1.

---

### Official Review · Reviewer_DL64 · 2024-07-30

**Soundness:** 2
**Presentation:** 3
**Contribution:** 2
**Rating:** 4
**Confidence:** 4

**Summary:**

The paper suggests the idea that reshuffling the splits for every configuration during the hyperparameter optimization can improve the generalization property. The paper derives the theoretical analysis to shows how reshuffling affects the asymptotic behaviour of the validation loss surface, and the paper also provides a bound on the expected regret in the limiting regime. Finally, the paper also conducts some experiments to demonstrate the effectiveness of the proposed approach.

**Strengths:**

+ The general idea of the paper is to reshuffle the sampling split when evaluating the hyperparameters, which is very interesting to me. It also makes sense and is reasonable.
+ The writing of the paper (except the theoretical analysis part) is good in general. I can understand the key ideas and also other details of the proposed approach.
+ The experiments include different aspects like understanding the shape of the loss function, and also it shows the improvement of the proposed approach when finding the optimal objective function values when applying to a state-of-the-art BO method.

**Weaknesses:**

Firstly, I have various concerns with the theoretical analysis conducted in the paper, in particular, I think they’re not rigorous. I list in the below some of my concerns regarding the theoretical analysis.
+ In Theorem 2.1, the “regularity conditions” need to be explicitly mentioned in the theorem (cannot be put in the appendix) because the theorem itself needs to be self-contained in the main paper.
+ In Theorem 2.1, it’s unclear whether the covariance matrix \Sigma exist (because it’s a limitation of an expression and what if that expression goes to infinity?). Furthermore, even when it exists, how is it related to n? If it is larger than O(n) then the theorem doesn’t have any meaning because it doesn’t show the convergence of \hat{\mu}(\lambda_j)?
+ In the discussion of Theorem 2.1, I don’t understand why we have Eq. (1). More explanations need to be provided.
+ In Theorem 2.2, why can we assume that \hat{\mu} follows a Gaussian process model? Is this assumption used in existing literature? And does it make sense in practice? Furthermore, what are the limits of the terms B(\tau) and A(\tau) in the RHS of the equation in Theorem 2.2? Without any quantification of these terms, we cannot conclude anything about how far \mu(\hat{\lambda}) from \mu(\lambda^*)

Secondly, for the experiments, my main concern is that the paper only evaluates the performance of the reshuffling approach using one state-of-the-art BO method (HEBO). I think it should be evaluated on at least 2 state-of-the-art BO methods to clearly demonstrate the effectiveness of the proposed approach.

**Questions:**

Please see my questions in the Weakness section.

**Limitations:**

The paper describes the limitations of their method in Section 5.

---

> ### Author Rebuttal · Authors · 2024-08-06
>
> Thank you very much for your helpful feedback on the presentation of our theoretic results, which will help us to improve their presentation. Below, we give detailed answers to your suggestions:
>
> **Concerns regarding the theoretical analysis**
>
> > In Theorem 2.1, the “regularity conditions” need to be explicitly mentioned in the theorem (cannot be put in the appendix) because the theorem itself needs to be self-contained in the main paper.
>
> Good point. We will surely do that.
> > In Theorem 2.1, it’s unclear whether the covariance matrix \Sigma exist (because it’s a limitation of an expression and what if that expression goes to infinity?). Furthermore, even when it exists, how is it related to n? If it is larger than O(n) then the theorem doesn’t have any meaning because it doesn’t show the convergence of \hat{\mu}(\lambda_j)?
>
> The covariance $\Sigma$ cannot diverge: First, the kernel $K$ is bounded because the loss function $\ell$ is assumed to be bounded (this could be relaxed by explicit conditions involving the generic learner $g_{\lambda}$, but we prefer not to overcomplicate matters here). Second, the term $\tau_{i, j, M}$ is bounded above by $1/\alpha^2$ because all probabilities are bounded by 1. The limit itself is, of course, no longer related to $n$ (as we have taken $n \to \infty$). We will add a corresponding remark to the paper.
>
> > In the discussion of Theorem 2.1, I don’t understand why we have Eq. (1). More explanations need to be provided.
>
> Please refer to our general comment, point 2.
>
> > In Theorem 2.2, why can we assume that \hat{\mu} follows a Gaussian process model? Is this assumption used in existing literature? And does it make sense in practice?
>
> We think this is an appropriate and realistic assumption:
> 1. Theorem 2.1 shows that this assumption at least holds in the large-sample limit, under standard assumptions (see, e.g., [1], [2]). Further, as stated in the sentence preceding Theorem 2.1:  *"This limiting regime will not only reveal the effect of reshuffling on the loss surface but also give us a tractable setting to study HPO performance."* This remark was intended to explain that a) the upcoming GP assumption is appropriate because it is approximately true due to Theorem 2.1, b) it makes a theoretical analysis of the generalization behavior possible. This sentence can certainly be made clearer.
> 2. From a state-of-the-art standpoint: Using a GP to model hyperparameter landscapes is one of the most successful HPO principles and very popular in BO. For example, the BO method HEBO we use in the experiments follows this approach. GPs are also directly used to model and analyze HP landscapes, for example in [3].
>
> [1] Bayle, P., Bayle, A., Janson, L., & Mackey, L. (2020). Cross-validation confidence intervals for test error. Advances in Neural Information Processing Systems, 33, 16339-16350.
> [2] Austern, M., & Zhou, W. (2020). Asymptotics of cross-validation. arXiv preprint arXiv:2001.11111.
> [3] Mohan et al., AutoRL Hyperparameter Landscapes. AutoML 2023.
>
> > Furthermore, what are the limits of the terms B(\tau) and A(\tau) in the RHS of the equation in Theorem 2.2? Without any quantification of these terms, we cannot conclude anything about how far \mu(\hat{\lambda}) from \mu(\lambda^*)
>
> The terms $B$ and $A$ have an explicit form that can be evaluated for a given data-generating process and hyper-parameter grid. The term $B(\tau)$, as stated, is unbounded, but a closer inspection of the proof shows that it is upper bounded by $\sqrt{\log J}$ (l661: *“we can use the trivial bound $N (s) \le J$”*). This bound is attained only in the unrealistic scenario when the validation losses are essentially uncorrelated across all HPCs. The term $A(\tau)$ is lower bounded by $0$, which is also the worst case because $A$ enters with a negative sign. We shall include this discussion in the revised version since it may also interest other readers.
>
> Finally, let us emphasize that, like most bounds in learning theory, this bound is not meant to provide a sharp quantitative result but to gain general insights about the influence of the various parameters on the generalization performance. The simulation study in the following section corroborates these insights.
>
> **Concerns regarding the experimental evaluation**
>
> Please refer to our general comment, point 1.

---

### Author Rebuttal · Authors · 2024-08-06

We thank all reviewers for their time, constructive feedback, useful suggestions, and positive evaluation. We want to use the space here to address two points that were raised by multiple reviewers.

**1. (DL64, sYf9, QU3x, rcFG) Additional experiments**

Some reviewers suggested various additional experiments. We value strong empirical analyses as much as theoretical insights, but we also argue that our paper already contains a very thorough experimental setup.
Summarizing what was said in Appendix E.4: We ran RS, BO (a modern and advanced version with a proven track record), 3 different version of resampling, and many learners which are arguable in the set of "SOTA methods for tabular data", on 10 different data sets with diverse characteristics. The RS experiments covered 33.6 Mio. model fits, for HEBO it was 24 Mio. We deem that "quite large scale". In view of the already extensive experiments involving many different data sets and dozens of millions of model fits, we have deliberately limited our scope to tabular data. We strongly believe that these experiments reflect realistic settings encountered by practitioners many times, see for example [vBvdS, BLSH+, MKVC+].

Because two reviewers were specifically interested in results for a second state-of-the-art HPO method to more clearly show the effectiveness of reshuffling, we have nevertheless conducted extra experiments in the interest of scientific discussion here. We ran SMAC (SMAC4HPO facade, [LEFB+]) in addition to HEBO on all 4 models (CatBoost, XGBoost, funnel-shaped MLP, Elastic Net) on all 30 tasks (3 train_valid sizes * 10 DGPs) with 10 repetitions for the standard and reshuffled holdout and provide an updated Figure 5 from the main submission in the uploaded rebuttal pdf. We can observe that reshuffling the holdout has a similar effect on SMAC as on HEBO and results in better generalization performance. We will continue running SMAC for all other resamplings, similarly as already reported for HEBO in Appendix F.1.


**2. (DL64, 8FSs) Confusion about Eq (1)**

There was some confusion about Eq (1) in the paper and where it comes from. We apologize for jumping too fast. The term $\tau_{i, j, M}$ appears in Theorem 2.1 and captures the influence of reshuffling on the loss surface.
Eq (1) follows from our method-specific computations for the term $\tau_{i, j, M}$ provided Appendix D. In all cases, the result happens to have the form displayed in Eq (1). The method-specific values for $\sigma$ and $\tau$ are then summarized in Table 1.

In the revised version, we will
* first introduce the different resampling methods,
* then refer to Appendix D for mathematical computations,
* only then state and explain Eq (1) and the values in Table 1;
* revise the presentation in Appendix D.2 to make the form appearing in Eq (1) more clearly visible.

We hope that this clarifies our paper and makes our results more accessible.

---
[vBvdS] Position: Why Tabular Foundation Models Should Be a Research Priority. Boris Van Breugel, Mihaela van der Schaar; ICML 2024.
[BLSH+] Deep neural networks and tabular data: A survey. Vadim Borisov, Tobias Leemann, Kathrin Seßler, Johannes Haug, Martin Pawelczyk, Gjergji Kasneci; IEEE 2022
[MKVC+] When Do Neural Nets Outperform Boosted Trees on Tabular Data? Duncan McElfresh, Sujay Khandagale, Jonathan Valverde, Vishak Prasad C, Ganesh Ramakrishnan, Micah Goldblum, Colin White, NeurIPS 2023
[LEFB+] SMAC3: A versatile Bayesian optimization package for hyperparameter optimization. Lindauer, M., Eggensperger, K., Feurer, M., Biedenkapp, A., Deng, D., Benjamins, C., ... & Hutter, F. Journal of Machine Learning Research, 23(54), 1-9, 2022

---

### Comment · Area_Chair_Z2yM · 2024-08-10
**Author Rebuttal Availible**

Dear Reviewers,

Thank you for your reviews and your time.  The authors have provided a rebuttal, answering specific questions that were raised and have also provided additional empirical observations.

Please take a careful look and follow up with additional questions and/or update your review as necessary. At the minimum indicate you have read and understood the feedback.

Also, please feel free to start a private discussion in a separate thread.

Thank you,
Area Chair for Submission 1124

---

### Decision · Program_Chairs · 2024-09-25

**Decision:**

Accept (poster)

**Comment:**

After the initial review, rebuttal, and discussion, the reviewers remain split on their decisions, but I am inclined to recommend accepting the submission.  Given its novel theoretical insights and thorough empirical analysis of a simple yet not well studied modification to ubiquitous hyperparameter optimization methodologies, the results in this work can provide value to both theoreticians and practitioners.

As reviewers pointed out (and authors acknowledged in the rebuttal) the clarity of presentation should be improved in several places.  I additionally suggest giving a few concrete examples of learning algorithms / model families that meet the regularity conditions required for your analysis. The additional empirical results using SMAC should also be included in the final version.

Finally, as mentioned in the thoughtfully written Limitations section, the theoretical analysis applies only to the setting in the limit of infinite data. Any insights on the possibility of (or obvious roadblocks) to a finite sample analysis would be a valuable addition and helpful for future extensions of this work.